# Structural landscape of the respiratory syncytial virus nucleocapsids

Lorène Gonnin[1,2], Ambroise Desfosses [1] ✉, Maria Bacia-Verloop[1], Didier Chevret[2], Marie Galloux[2], Jean-François Éléouët[2] & Irina Gutsche [1] ✉

Human Respiratory Syncytial Virus (HRSV) is a prevalent cause of severe respiratory infections in children and the elderly. The helical HRSV nucleo-capsid is a template for the viral RNA synthesis and a scaffold for the virion assembly. This cryo-electron microscopy analysis reveals the non-canonical arrangement of the HRSV nucleocapsid helix, composed of 16 nucleoproteins per asymmetric unit, and the resulting systematic variations in the RNA accessibility. We demonstrate that this unique helical symmetry originates from longitudinal interactions by the C-terminal arm of the HRSV nucleoprotein. We explore the polymorphism of the nucleocapsid-like assemblies, report five structures of the full-length particles and two alternative arrangements formed by a C-terminally truncated nucleoprotein mutant, and demonstrate the functional importance of the identified longitudinal interfaces. We put all these findings in the context of the HRSV RNA synthesis machinery and delineate the structural basis for its further investigation.

Human respiratory syncytial virus (HRSV) is the most frequent cause of bronchiolitis and pneumonia in infants and a major cause of childhood death in low-income settings[1,2]. Reinfection can occur throughout life and is often serious in elderly and immunocompromised. First vaccines for older adults have just been licenced[3,4], several options for pregnant women are in the pipeline, and the terms and conditions of the usage of a new generation of monoclonal antibodies preventing severe HRSV infections in babies are currently being worked out[4]. There are however still no therapeutic options available for the vast majority of the population including vulnerable patients at risk of severe infections, with treatment limited to supportive care. Development of effective therapeutics requires a better understanding of the HRSV synthesis machinery. HRSV belongs to the *Mononegavirales* order with the non-segmented negative-strand RNA genome fully coated by the viral nucleoprotein N. The resulting helical nucleocapsid (NC) shields the viral genetic material from recognition by the innate immune system while serving as template for replication and transcription by the viral RNA polymerase complex, thereby constituting a potential drug target.

Alongside HRSV and human Metapneumovirus (HMPV), belonging to the *Pneumoviridae* family, *Mononegavirales* contains other important human pathogens such as the *Rhabdoviridae* rabies, the *Filoviridae* Ebola (EBOV) and Marburg (MARV), and the *Paramyxoviridae* measles (MV), mumps (MuV) and Nipah (NiV) viruses. *Pneumoviridae* are equally distant to *Paramyxo-* and *Filoviridae*[5]. In particular, as far as the NCs are concerned (i) each paramyxo- and filoviral N binds precisely 6 nucleotides, whereas pneumoviral N binds[6,7]; (ii) the genome size of paramyxo- but not pneumo- and filoviruses is a strict multiple of 6 nucleotides; (iii) paramyxo- and filo- but not pneumoviral genomes require bipartite promoters separated by an exact multiple of 6 nucleotides[7]; (iv) paramyxo- and filoviral N possess a very long C-terminal extension involved in replication and transcription, whereas the pneumoviral N features only a short C-terminal arm (i.e., the length of MV N is 525, EBOV NP 739 and HRSV N 391 amino acids respectively)[8]. Removal of the C-terminal extension rigidifies and condenses the helical paramyxo- and filoviral NCs by strengthening the contacts between successive turns, thus facilitating their structural analysis by cryo-electron microscopy (cryo-EM) and tomography (cryo-ET).

Despite a recent massive increase in the number of medium and high-resolution cryo-EM structures of the helical paramyxo- and filo-viral NCs[9–18], a detailed cryo-EM characterisation of the pneumoviral

[1]Institut de Biologie Structurale, Univ Grenoble Alpes, CEA, CNRS, IBS, 71 Avenue des martyrs, F-38044 Grenoble, France. [2]VIM, Paris-Saclay University, INRAE, 78350 Jouy-en-Josas, France. ✉e-mail: ambroise.desfosses@ibs.fr; irina.gutsche@ibs.fr

NCs is still lacking. Here we present an exhaustive cryo-EM analysis of the structural landscape of HRSV NCs in solution. We reveal in particular the non-canonical helical symmetry of the HRSV NC, with 16 nucleoproteins per asymmetric unit, and demonstrate that this unique organisation results from inter-turn interactions by the C-terminal arm of N and leads to periodic variations in the RNA accessibility along the NC filament.

## Results

### HRSV nucleocapsids are flexible and polymorphic

Earlier negative stain and cryo-negative stain EM studies reported polymorphism of HRSV N-RNA assemblies observed as flexible filaments and rings[6,19,20] upon virion lysis or by heterologous expression of N. The current structural information about HRSV NCs comes mostly from the 3.3 Å resolution X-ray crystal structure of decameric N-RNA rings[6] ($N_{10}$ ring, PDB: 2WJ8), a negative stain electron tomography analysis of purified helical NCs[21] and two cryo-ET studies of the HRSV virion[22,23]. Our cryo-EM images of recombinant HRSV N purified from insect cells displayed a polymorphic ensemble in which ring-like particles and filaments could be distinguished and classified (Fig. 1a, b). In addition to the well-known C10- and C11-symmetric rings[6,19,20] with a strongly predominant top view orientation, a significant amount of side views corresponding to two decameric rings stacked bottom-to-bottom could be detected. Since this assembly, hereafter termed $N_{10}$ double ring, had not been previously described, its 3D cryo-EM map was derived from the corresponding side view classes. In parallel, the filaments were split into sets of classes showing either continuous or discontinuous course. The former was used for 3D reconstruction of a helical NC and its ~1.5-turn subsection, whereas the latter yielded

reconstructions of a double-headed NC and a ring-capped NC. Thus, five different 3D maps—a double ring, a helical NC and its short subsection, a double-headed NC and a ring-capped NC—were obtained from the same dataset (Fig. 1c–g; Supplementary Fig. 1).

### Unique tripartite stabilisation of the HRSV N oligomerisation inside a conserved N-hole

The 2.86 Å resolution map and the resulting atomic model of the $N_{10}$ double ring show that the N protomer and the entire $N_{10}$ ring are identical to the crystal structure, with 0.5-Å RMSD over 378 backbone residues and the density of the last twelve residues (380–391) largely disordered. Accordingly, the RNA binding groove formed by the interface between the N-terminal and the C-terminal domains of N (NTD and CTD), and the "4-bases-in, 3-bases-out" RNA conformation remain unaltered.

Similarly to other *Mononegavirales*[24,25], the N- and C-terminal extensions of HRSV N, termed NTD-arm (residues 1–36) and CTD-arm (residues 360–391) (Fig. 1h), interact with the laterally adjacent N protomers thereby stabilising their oligomeric assembly on the RNA strand by subdomain swapping (Fig. 2a). The visible part of the CTD-arm of $N_i$ lies on top of the CTD of $N_{i+1}$ implying that in a helical NC it should be situated in between consecutive turns[6]. In parallel, the NTD-arm of $N_{i+1}$ inserts into a compact fold of the CTD of $N_i$ from the ring interior and extends towards the CTD-arm of $N_{i-1}$. In this regard, a "latch-bolt type" interaction formed by an insertion of a loop from the NTD of $N_{i-1}$ into an $N_i$ cavity, termed N-hole, has been recently described for paramyxoviral NCs[14,15,25], and is also present in filoviral NCs (Supplementary Fig. 2). The structures of HRSV and HMPV $N_{10}$ rings[26] (PDB: 5FVC) indicate that pneumoviral NCs do actually possess

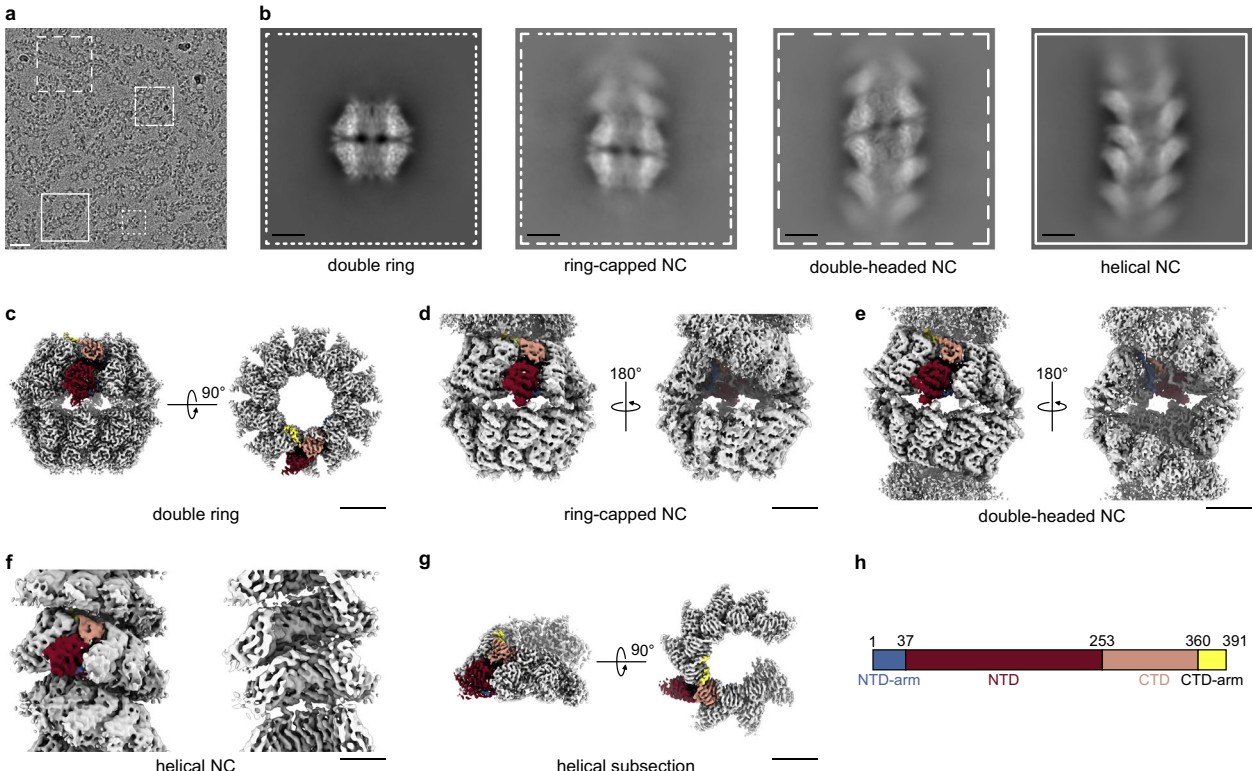

**Fig. 1 | Cryo-EM analysis of HRSV NCs. a** A representative micrograph of HRSV NCs purified from insect cells and featuring double rings, ring-capped NCs, double-headed NCs and helical NCs. 11,386 micrographs were selected for further processing. Particles are boxed, as an illustration, scale bar 200 Å. **b** Representative 2D classes with the outline matching the particles highlighted in **a**. **c** Cryo-EM map of the $N_{10}$ double ring (*EMD:17031*, side and top view). **d** Cryo-EM map of the ring-capped NC (*EMD:17037*, front and back view). **e** Cryo-EM map of the double-headed NC (*EMD:17036*, front and back view). **f** Cryo-EM map of the helical NC (*EMD:17030*, front and cut-through view). **g** Helical subsection (*EMD:17035*, side and top view). Scale bar, 50 Å in **b**–**g**. **h** Schematic of the HRSV N sequence divided into an NTD-arm (blue-grey), NTD (rosewood), CTD (old rose) and CTD-arm (yellow). In cryo-EM maps in **c**–**g**, one protomer is coloured according to this schematic, with the RNA in black.

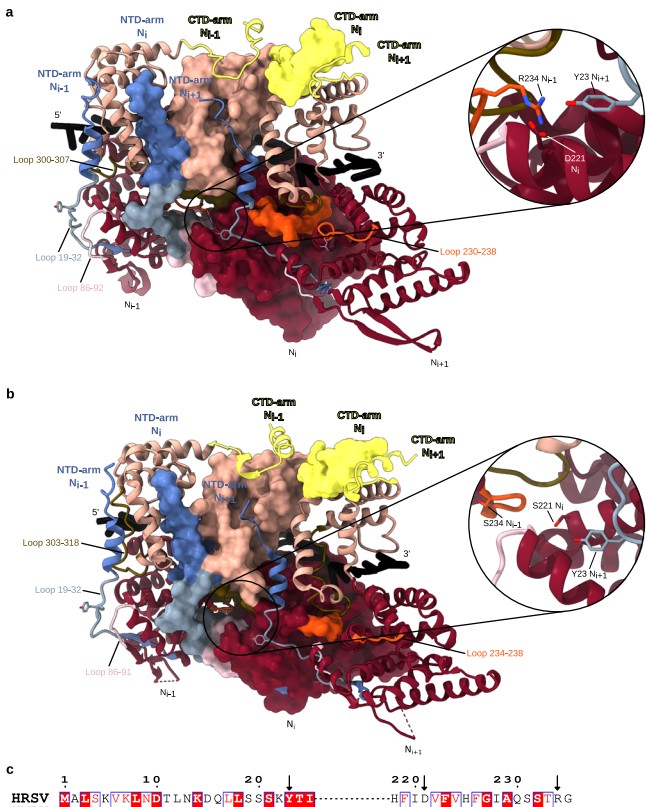

**Fig. 2 | Lateral interactions between N protomers in HRSV and HMPV N$_{10}$ double rings. a** Atomic models of three consecutive HRSV N protomers are shown, the middle one as a surface and the edge ones as ribbons. NTD-arm, NTD, CTD and CTD-arm coloured as in Fig. 1, loop 19–32 in powder blue, loop 86–92 in mimi pink, loop 230–238 in orange and loop 300–307 in olive. The close-up of the N-hole of the middle protomer shows the tripartite Y23-D221-R234 interaction. **b** Same as **a** but for three protomers from the HMPV N-RNA ring crystal structure (PDB: 5FVC). Colouring as in Fig. 1 and in **a**, loop 19–32 in powder blue, loop 86–91 in mimi pink, loop 234–238 in orange and loop 303–318 loop in olive. The close-up shows the absence of a tripartite interaction in the N-hole. **c** Pairwise sequence alignment of HRSV and HMPV N around the residues involved in the tripartite interaction in the HRSV N oligomer. Conserved residues in red boxes; arrows pointing at residues 23, 221, and 234.

a cognate N-hole formed by an extended NTD-arm-proximal loop (residues 19–32 in HRSV N), together with two short loops from the NTD (86–92) and the CTD (300–307). Likewise, in HRSV and HMPV rings, a short loop from the NTD of N$_{i-1}$ (residues 230–238 in HRSV N) protrudes into the N-hole of N$_i$ (Fig. 2a, b), which demonstrates that the N-hole-based interaction is conserved between *Paramyxo-*, *Filo-* and *Pneumoviridae* families (Supplementary Fig. 2).

Deeper into the N-hole matter, the first atomic model of an HRSV NC helix (PDB: 4BKK), derived from a crystal structure fit into a tomography-based featureless 68-Å pitch spiral, suggested a fascinating direct interaction between three consecutive protomers[21]. Specifically, R234 of N$_{i-1}$ was predicted to bind both D221 of N$_i$ and Y23 from N$_{i+1}$. Such a tripartite interaction between N$_{i-1}$, N$_i$ and N$_{i+1}$ does not exist in paramyxo- and filoviral NCs, and to our knowledge has not been explicitly investigated for the HMPV N$_{10}$ ring. Examination of our HRSV double ring structure verifies the presence of the tripartite Y23-D221-R234 interaction and shows that it occurs inside the N-hole of N$_i$ which carries D221; the loop 230–238 of N$_{i-1}$ provides R234 while Y23 is contributed by the loop 18–32 of N$_{i+1}$, whereas Y23 on the equivalent loop of N$_i$ points into the N-hole of N$_{i-1}$ so that to bind R234 of N$_{i-2}$ (Fig. 2a). Surprisingly, despite a great resemblance between the HRSV and the HMPV N-RNA rings, the latter contains no tripartite contact

(Fig. 2b). Indeed, although all loops are in place and Y23 is conserved, in HMPV N both D221 and R234 are replaced by serines making the interaction impossible (Fig. 2c). Thus, an additional tripartite stabilisation of the "latch-bolt type" interaction seems to be a signature of the HRSV NCs.

## Molecular determinants of the longitudinal NTD-NTD interaction

In the double ring, the NTD-NTD stacking of two N$_{10}$ rings, whose centres of gravity are 67 Å apart, is assured by D1-symmetry-related β-sheets providing two opposing interacting H100 residues and two R101-E122 hydrogen bonds (Fig. 3). Interestingly, examination of the crystal structures of HRSV and HMPV rings (Fig. 3a–c; Supplementary Fig. 3) reveals their bottom-to-bottom (NTD-NTD) stacking but with a tighter packing, with an inter-ring distance of 61 Å and 60 Å respectively. This compaction arises from an inter-ring rotation accompanied by a β-sheet insertion into inter-protomer grooves of the opposite ring (Fig. 3a–c), which leads to a difference between the crystallographic inter-ring interface, based on a K91-D96 interaction, and the solution one.

2D classification of segments of filamentous HRSV NC produced some 2D class averages featuring a clear seam, either across the filament stem or close to its end (Fig. 1b). Particles with the stem-crossing seam yielded a 3.9 Å resolution map with a barbed end-to-barbed end junction of two NC helices (Fig. 1e; Supplementary Fig. 1), similar to the spiral clams described for *Paramyxoviridae* Sendai (SenV)[15], NiV[27] and Newcastle disease (avian paramyxovirus 1 or APMV-1)[12] viruses. The particles with an end-proximal seam gave a 3.8 Å resolution map of a helical NC capped by an N$_{10}$ ring (Fig. 1d; Supplementary Fig. 1), reminiscent of the semi-spiral clam observed for NiV NCs[27]. Remarkably, the mode of the longitudinal NTD-NTD interaction in the double rings, the double-headed and the ring-capped HRSV NCs is conserved (Fig. 3a, d, e), confirming that the interface delineated by cryo-EM is more reflective of the native structures than the crystal structure interface constrained by the crystal packing. In HRSV, the NTD-NTD interface is however distinct from the one in the NiV, SenV and APMV-1 clams, mediated by NTD loops which are absent in pneumoviral N (Supplementary Fig. 4).

All *Mononegavirales* NCs are left-handed helices, with the CTDs and the 3'-end of the RNA oriented towards the pointed end of the filaments and the NTDs and the 5'-end towards the barbed end[24]. The paramyxoviral clam-shaped assemblies were proposed to seed the growth of the double-headed helices from the 5' to the 3' end, protect the 5' end from nucleases[12] and support encapsidation of several NCs per virion[28], also documented for HRSV[23,29]. Thus, based on our structures, we designed two double mutants of N—H100E-R101D and H100E-E122R—and assessed their phenotypes in an HRSV minigenome assay. While the first construct behaved similarly to the wild type N, the H100E-E122R mutation resulted in a circa 90% reduction of the polymerase activity (Supplementary Fig. 5), which suggests a possible functional role of the NTD-NTD interactions in the HRSV RNA synthesis.

## Cryo-EM analysis reveals a non-canonical symmetry of the helical HRSV NC

Although at first glance, the 2D class averages of the HRSV NCs with a continuous filament course suggest a paramyxoviral-like arrangement with a herringbone appearance and a ~70 Å pitch, their careful scrutiny shows that in all class averages every ~1.5 turns (or ~100 Å) densities at either the left- or the right-hand side of the pattern are shifted inwards (Fig. 1b; Supplementary Fig. 6). The power spectrum (PS) of the 2D classes exhibit an expected layer line with the maximum close to the meridian at ~1/70 Å, attributable to the estimated pitch. Surprisingly however, the PS also features an additional layer line, with a strong maximum on the meridian, at ~1/100 Å, pinpointing a periodicity that should correspond to a ~100 Å rise (Supplementary Fig. 6). Since

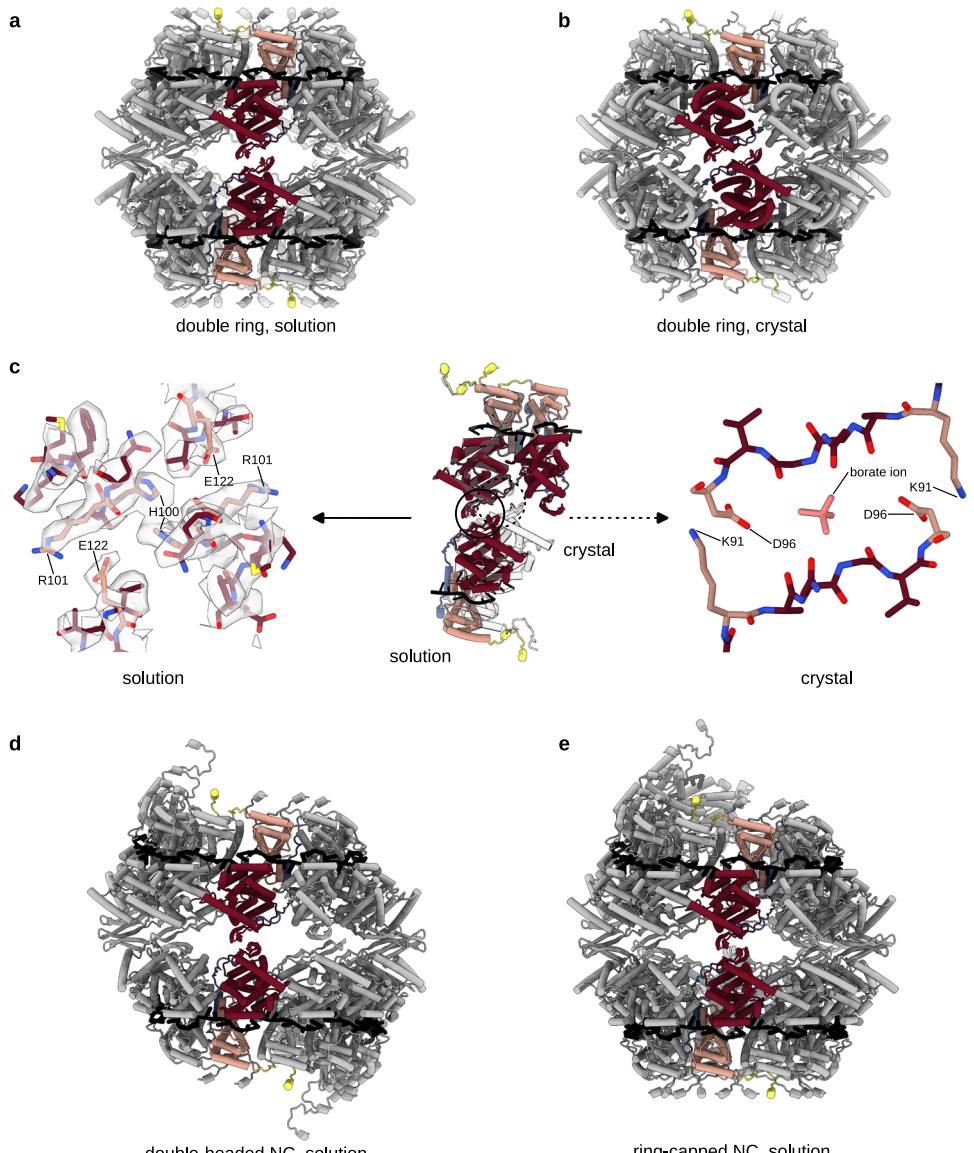

**Fig. 3 | Longitudinal NTD-NTD interactions conserved between HRSV N$_{10}$ double ring, double-headed and ring-capped NCs but different from the crystal structure of the HRSV N$_{10}$ double ring.** In each panel, two opposite protomers are coloured as in Fig. 1. **a** Atomic model of the HRSV N$_{10}$ double ring derived from the cryo-EM map shown as cartoon. **b** Atomic model of the HRSV N$_{10}$ double ring crystal structure (PDB: 2WJ8). **c** Alignment of the top rings of the cryo-EM and crystal structure-based models reveals a rotation between the bottom rings. Two top-ring protomers and one opposing bottom-ring protomer are shown in the middle of the panel, with the cryo-EM-based structure coloured as in Fig. 1 and the crystal structure in white. A close-up of the cryo-EM map and the atomic model highlighting the NTD-NTD interactions is on the left, a close-up of the NTD-NTD interactions in the crystal structure is on the right. The difference between the crystallographic and the solution inter-ring interfaces may be related to the presence of a borate ion in the interaction site in PDB: 2WJ8, possibly embarked during the electrophoretic separation of decameric and undecameric HRSV N-RNA rings prior to crystallisation. **d** Atomic model of the double-headed NC. **e** Atomic model of the ring-capped NC.

geometrically the rise cannot be larger than the pitch, this implies that the measured value of the rise does not reflect the axial shift between two consecutive protomers. In principle, the ~100 Å periodicity could arise from stacking of short ~1.5-turn helices with a 70 Å pitch; however, no discontinuity and no isolated ~1.5-turn helices were observed in our cryo-EM images despite exhaustive particle picking and extensive 2D classification. Alternatively, if the NCs are continuous, they would be organised in ~1.5-turn asymmetric units composed of multiple N protomers.

3D reconstructions with a 100 Å rise as a starting value and a variable twist led to a solution with correct secondary structures of N, and a subsequent isolation of the straightest NCs yielded a final 3D map at an average resolution of 6.2 Å and a continuous RNA density (Fig. 1f; Supplementary Fig. 1; Fig. 4; Supplementary Fig. 6). This

moderate resolution lies in the short-range order of the helical HRSV NC. Indeed, an additional 3D refinement within a mask enclosing ~1.5 turns resulted in a 3.5 Å average resolution map of a five protomer-subsection in the middle of the mask, which however rapidly deteriorates towards the mask periphery due to a progressive loss of regularity (Fig. 1g; Supplementary Fig. 1). The structure of the N-RNA protomer is again largely the same as in the crystal, with RMSD less than 1 Å over 378 backbone residues for each of the five protomers, and the inter-protomer contacts maintained.

The determined helical parameters and an inspection of the map and the model of the NC helix (Fig. 1f; Supplementary Fig. 1; Fig. 4a, b) allow to interpret the peculiar experimental class averages and the PS (Supplementary Fig. 6). Indeed, the HRSV NC reveals itself as a right-handed "super-helix", defined by a 105.3 Å rise and a 149.5° twist,

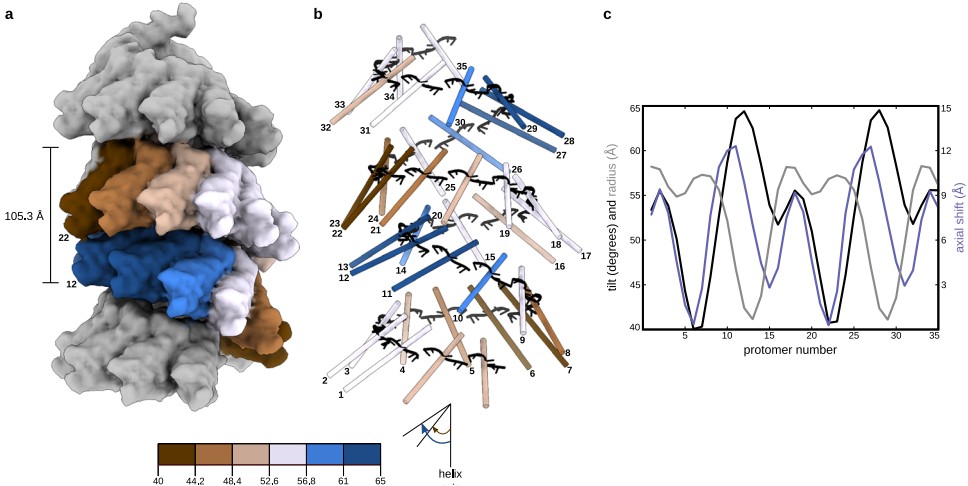

**Fig. 4 | Non-canonical helical symmetry of the HRSV NC. a** Atomic model of the NC is filtered to 10 Å resolution and displayed as surface. Protomers in one asymmetric unit are coloured dependent on their axial tilt following the colour code shown at the schematic underneath, the rest of the protomers are coloured in grey. **b** Protomers of the model in **a** are shown as sticks coloured dependent on the protomer axial tilt and numbered 1 to 35. **c** Plot showing the axial tilt (black), the radial position (grey) and the relative axial shift of each protomer.

generated by helical repetition of asymmetric units composed of 16 N protomers forming a ~1.5 turns left-handed spiral staircase. Inside each asymmetric unit, the protomer arrangement is similar to that observed in paramyxoviral helical NCs. Amazingly however, the position and the orientation of the protomers relative to the filament axis as well as the axial shift between two consecutive protomers undergo a specific and coordinated variation (Fig. 4c). For example, the tilt of the protomers varies between ~40° for the most "standing" ($N_6$ and $N_7$) and ~65° for the most "lying" ($N_{11}$, $N_{12}$, and $N_{13}$), whereby the most lying subunits are the closest to the helical axis. The combination between the helical parameters and the variation profile of the protomer poses in the asymmetric unit engenders an axial alternation of regions where two neighbouring turns are the closest to each other and regions where they are spread further apart. This alternation occurs circa every 105 Å/ 1.5 turns/16 protomers and manifests itself by clamping the helix on a side, thereby increasing the gaps above and below the clamps. The helical propagation of the clamps and gaps confers the HRSV NC its unique appearance, accentuated by an inward shift of the densities corresponding to projections of the most lying subunits in the 2D class averages (Fig. 4; Supplementary Fig. 6). The variation of the protomer tilt is visible even on the five protomer-subsection of the helix (Supplementary Fig. 1). The numbering of the protomers in the asymmetric unit is done based on the correspondence between their axial tilts in the double-headed NC, where the first barbed-end subunit is clearly identified, and in the helical NC (Supplementary Fig. 7).

### Periodic variations of RNA accessibility and the CTD-arm-mediated inter-turn interactions in helical HRSV NCs

One consequence of this NC organisation is particularly conspicuous: in the configuration where the lower-turn protomers are lying and shifted inwards and the upper-turn protomers standing above, the RNA of the lying protomers is hidden inside the clamps; in contrast, in the standing-protomer configuration the RNA appears exposed (Fig. 5). Another striking observation (Fig. 5) is that three consecutive "nearly standing" protomers in the lower turn interact with the upper turn through their CTD-arms, which are therefore better defined than in the other protomers where they are not constrained. Indeed, inspection of the CTD-arm densities in the five protomer-subsection and in the entire asymmetric unit shows that the definition of the CTD-arm of each protomer $N_i$ depends on the position and orientation of the protomer(s) located immediately above (i.e., $N_{i+10}$, $N_{i+11}$ and $N_{i+12}$) (Fig. 5). Although modelling of the CTD-arm after the residue 379[6]

would be unreliable, a rigid body fit into the subsection map indicates that the three "nearly standing" subunits of the helical HRSV NCs do show densities extending beyond. The CTD-arms of the subunits $N_2$ and $N_3$ (equivalent to $N_{18}$ and $N_{19}$ in Figs. 4 and 5) seem to contact almost the same zones in the subunits $N_{13}$ and $N_{14}$, (i.e., $N_{29}$ and $N_{30}$) respectively, whereas the CTD-arm of the subunit $N_4$ (*i.e.* $N_{20}$) falls nearly in between the upper subunits $N_{14}$ and $N_{15}$ (i.e., $N_{30}$ and $N_{31}$) because of the singular helical symmetry of the HRSV NC (Fig. 5).

### Shortening of the CTD-arm transforms the HRSV NCs into paramyxoviral-like canonical helices

Since the structure of the helical HRSV NC demonstrates the involvement of the CTD-arm in the longitudinal contacts, we supposed that shortening of this arm may abrogate inter-turn interactions and thereby transform the non-canonical helix with asymmetric units composed of 16 N arranged in ~1.5 turns into a classical helix with one N per asymmetric unit (Supplementary Fig. 6; Supplementary Movie 1). Considering the previously published data on the major role played by the last 20 residues of N in the HRSV polymerase activity and the critical requirement of the N residue L370 for the stabilisation of the $N^0P$ complex[30], we opted for a C-terminally truncated N1-370 construct. As expected from the ealier studies[30], the CTD-arm truncation completely abolishes the polymerase activity in an HRSV minigenome assay (Supplementary Fig. 8), highlighting the importance of the CTD-arm of N in the HRSV RNA synthesis. In spite of the truncation, the N1-370 mutant retained its interaction with P and therefore the corresponding N-RNA assemblies could be purified following the same protocol as the one used for the full-length (FL) construct and analysed by cryo-EM. The most glaring differences with the cryo-EM images of the full-length (FL) N-RNA were the appearance of two new types of coexisting filaments—the herringbone-like helices and a major population of unforeseen rigid stacks of rings (Fig. 6; Supplementary Fig. 1). The class averages and the PS of the helical NC formed by the N1-370 mutant are similar to those of paramyxoviral NCs, and the helical parameters of the resulting 4.3 Å resolution 3D map, 6.58 Å rise and −36° twist, closely agree with the ones derived from the FL "super-helix" (Supplementary Fig. 6). Moreover, the mutant manifests no longitudinal contacts (Fig. 6a–c), validating the structure-based hypothesis that it is the CTD-arm of the HRSV N that, by periodically linking two successive helical turns, induces the non-canonical symmetry of the HRSV NC and the resulting systematic variations in the RNA accessibility.

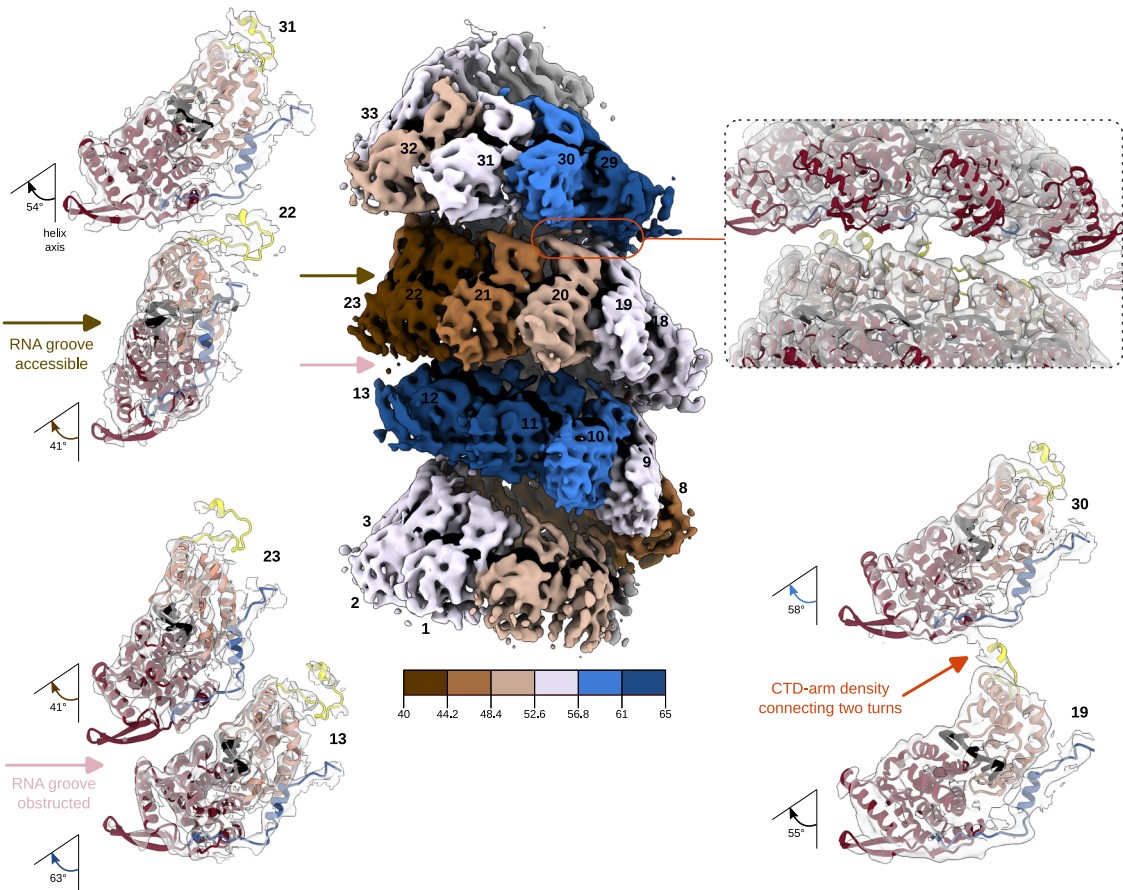

**Fig. 5 | RNA accessibility and CTD-arm-mediated inter-turn interactions in the helical NC.** Cryo-EM map of the helical NC before further refinement of the asymmetric unit (EMD:17030) is shown in the middle, coloured as in Fig. 4 and as reminded in the schematic underneath the map, with protomers numbered as in Fig. 4. On the left, close-up views of two sets of opposing protomers from two successive helical turns are shown to illustrate the difference in the RNA accessibility, with the cryo-EM density in transparent grey and the atomic model represented as a ribbon and coloured as in Fig. 1. On the top right, a similarly-coloured view of the inter-turn interaction is shown to highlight the densities corresponding to the CTD-arm, with a corresponding two-protomer close-up underneath. RNA is in black.

The rigid polymers coexisting with these helices are D10-symmetric and formed by alternating bottom-to-bottom and top-to-top packing of $N_{10}$ rings (Fig. 6d–f). Thus, these stacks are very different from those observed for digested mumps N-RNA rings packed top-to-bottom[14]. The ensuing 2.8 Å resolution cryo-EM map shows that the NTD-NTD stacked units are indistinguishable from the $N_{10}$ double rings of FL N until the end of the α-helix 344–358 and the beginning of the CTD-arm. In the rings and helical assemblies of the FL N, the CTD-arm of the subunit $N_i$ protrudes straight onto the top of the CTD of $N_{i+1}$. However, in the stacked rings of the N1-370 mutant, the truncated CTD-arm sharply pivots away and, instead of engaging into a lateral interaction, tucks into an identical site but on a CTD of the opposite ring in the stack. The pivoting of all CDT-arms tightly locks the adjacent rings together through their CTDs, such as to generate a polymer built of layers of inversely oriented $N_{10}$ rings engaged both in NTD-NTD and CTD-CTD contacts; the latter are additionally stabilised by binding between CTD-arms of two opposing protomers, in particular through a Y365-Y365 stacking (Fig. 6g, h). The involvement of specific structural elements of N in different lateral and longitudinal interactions underlying the remarkable polymorphism of HRSV NC-like assemblies is schematically depicted in Supplementary Fig. 10.

## Discussion

The major finding of this work was the non-canonical helical organisation of the HRSV NC, generated by ~1.5-turn asymmetric units composed of 16 N protomers, which undergo a concerted variation of their poses while remaining in quasi-equivalent environments. This unique symmetry, together with the great flexibility of the NCs, complicates their high-resolution analysis, and is totally different from those described for other *Mononegavirales* NCs. Excitingly, the arrangement of the HRSV NC is reminiscent of the one proposed for the Dahlemense strain of tobacco mosaic virus (TMV), and may be similarly considered in terms of a periodic deformation of a regular helical structure[31]. In the Dahlemense TMV model, the additional meridional reflexions appear on the layer lines halfway between those of the common TMV and the asymmetric unit contains exactly two turns. Likewise, for the HRSV NC the maximum on the meridian is observed at two-thirds of the layer line of the expected pitch and the asymmetric unit contains ~1.5 turns. The exterior distortion proposed in the Dahlemense TMV model is explained by the inside and outside sets of inter-turn interaction being incompatible with the same periodicity, whereas the common TMV does not have any axial outside interactions. The C-terminally truncated HRSV NC mutant also has no inter-turn interactions and features a canonical helical symmetry with equivalent environments for each N protomer. In contrast, in the FL NC, the CTD-arms at the filament interior are periodically involved in axial interactions with the upper turn, which induces a global structural reorganisation leading to tilting and inwards shifting of certain protomers and manifesting itself as a helical distortion at the NC exterior. Continuing the parallel, the stability of the observed full-length HRSV NC structure would indicate that "the decrease in the free energy upon forming some additional bonds is greater than the

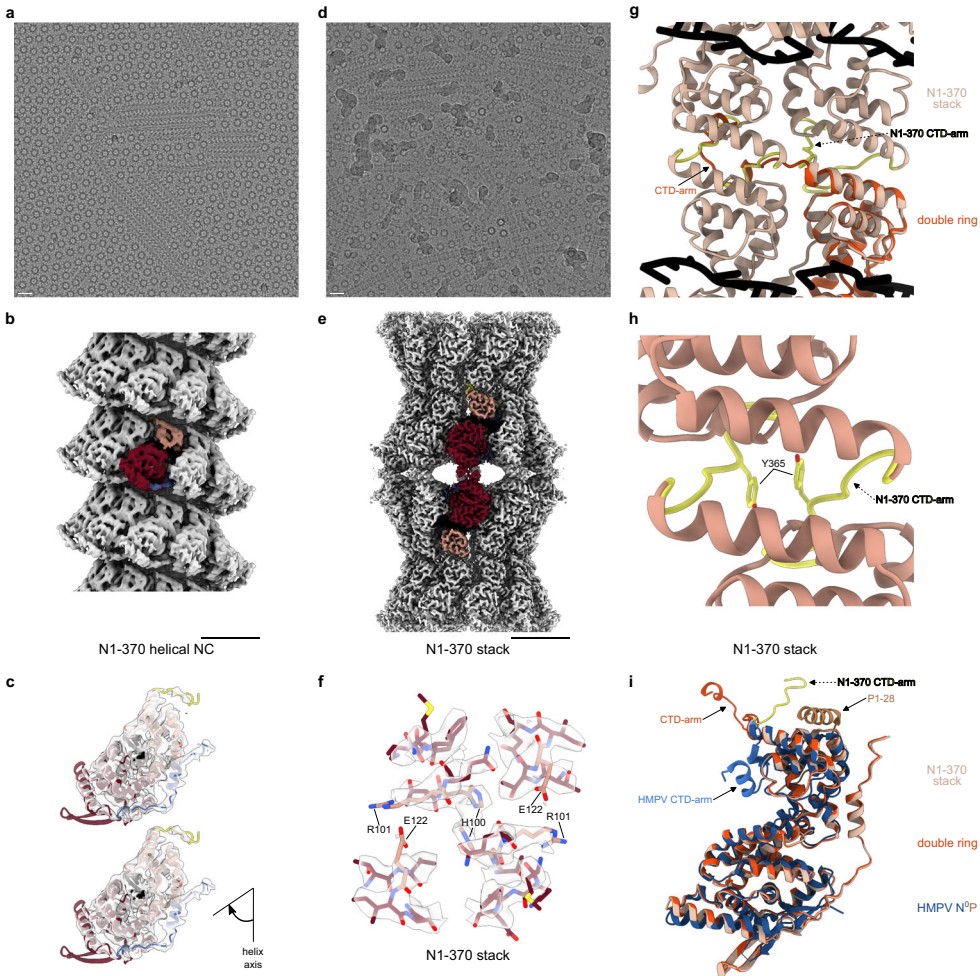

**Fig. 6 | Canonical helical NCs and stacked assemblies formed by the N1-370 mutant. a** A representative micrograph of the N1-370 NCs featuring mostly helical NCs and rings, scale bar 200 Å. 6312 micrographs were selected for processing. **b** Cryo-EM map of the canonical N1-370 helical NC (EMD:17034), with one protomer coloured as in Fig. 1, scale bar 50 Å. **c** Close-up view of protomers from two successive helical turns are shown to illustrate the position of the CTD-arm and the absence of inter-turn interactions, with the cryo-EM map in transparent grey and the atomic model represented as a ribbon and coloured as in Fig. 1. **d** A representative micrograph of the N1-370 NCs featuring helical NCs, stacks and rings, scale bar 200 Å. 6312 micrographs were selected for further processing. **e** Cryo-EM map of the N1-370 stack (EMD:17038), with one protomer coloured as in Fig. 1, scale

bar 50 Å. **f** Close-up of the cryo-EM map of the N1-370 stack and the atomic model highlighting the NTD-NTD interactions (similar to the ones in the $N_{10}$ double ring shown in Fig. 3c). **g** Alignment of the atomic models of the $N_{10}$ double ring and the N1-370 stack illustrating the difference in the orientation of the CTD-arms. One protomer of the $N_{10}$ double ring is shown in the bottom-right and coloured in orange, 4 protomers of the N1-370 stack are shown and coloured in beige, with the CTD-arms in yellow. RNA is in black. **h** Close-up of the atomic model of the N1-370 stack highlighting the CTD-CTD interactions. **i** Alignement of N protomers of the $N_{10}$ double ring (orange), the N1-370 stack (beige) and the HMPV $N^0P$ crystal structure ($N^0$ in blue and P1-28 in brown) (PDB: 5FVD). Positions of the CTD-arms are indicated.

increase in the free energy required to move the subunits into the slightly different, but quasi-equivalent positions"[31].

The helical NC, together with the RNA polymerase L, its phosphoprotein cofactor P and the transcription factor M2-1, form the HRSV RNA synthesis complex that constitues the minimal infectious unit of the virus. P acts as a central hub by tethering L to the NC template, chaperoning neosynthesised N such as to keep it monomeric and RNA-free ($N^0$) for specific nascent RNA encapsidation, and recruiting M2-1[32]. The matrix protein M is thought to direct HRSV assembly and budding by interacting both with the NC-bound P and with the envelope glycoprotein F[33,34]. Recent cryo-ET analysis of filamentous HRSV virions demonstrated that M is organised in a helical array that would coordinate helical ordering of glycoprotein spikes[23]. In addition, in *Paramyxoviridae*, M was shown to directly bind the CTD-arm of N[35,36]. Thus, HRSV M may potentially also influence the helical parameters of the NC upon the viral cycle via a direct or a P-mediated interaction. Noteworthy, the binding pocket of P on the NC helix is

situated far both from the CTD-arm itself and from its binding site to the upper helical turn[37,38].

In order to bind M or another viral or host factor, the end of the CTD-arm of N would need to escape outside the NC through an interstice between two turns, as shown for *Paramyxo-* and *Filoviridae*, which however have a much longer CTD extension. Here we showed that the non-canonical helical organisation of the HRSV NC is engendered by the last 20 residues of the CTD-arm of N. Considering the high similarity between HRSV and HMPV N, we suppose that all *Pneumoviridae* NCs adopt an analogous arrangement. Our structures and the structural homology between HRSV and HMPV $N^0P$ complexes inferred from biochemical studies[30] suggest that, similarly to the situation in HMPV[26], binding of the N-terminal peptide of HRSV P to $N^0$ would hamper its self-oligomerisation by preventing the CTD-arm from subdomain swapping and flipping it downwards along the core of N such as to block the RNA binding[26]. In contrast, in the stacked N1-370 rings, the truncated CTD-arm rotates upwards to dock into dedicated

subdomain swapping site on the opposite N protomer from the ring above. Thus, comparison of these structures indicates that the CTD-arm is able to explore a large angular space (Fig. 6i).

While the truncation of the CTD-arm is not supposed to occur in vivo, the high rotational freedom of the CTD-arm hints to the possibility of its reorientation upon interaction with viral or host factors. One may therefore envision a temporary capping of the pointed end of the NC helix by an N-RNA ring or a second NC through a CTD-CTD interaction, such as to protect the 3′ end of the RNA from host antiviral responses. This would however imply an uncapping of the 3′ end in order to initiate transcription or replication from the respective promoters that reside at the pointed end of the NC, by either P or L or another factor. The 5′ end, in its turn, would be protected inside the double-helical and 5′ ring-capped NCs formed through NTD-NTD interactions. Such top-to-top and/or bottom-to-bottom assemblies of HRSV NCs would be consistent with the observations in the HRSV virion[23,29] and in the infected cells[39].

The most obvious consequence of the non-canonical HRSV NC structure is the periodic variation of the RNA accessibility, the RNA binding groove of N being severely obstructed in the inwards-shifted lying protomers and exposed in the standing ones. While this difference in access to the RNA should inevitably influence pneumoviral synthesis by L during its gliding along the NC, the exact mechanistic implications of the observed variations are difficult to conceptualise. Indeed, because of the limited long-range order of the HRSV NC helices, any prediction of the protomer poses based on the numbering in the clam and semi-clam structures as adopted here can only be reliable for the protomers located very close to the barbed, 5′-end. Even an attempt to estimate the tilt of the protomer containing the most 5′-end-proximal gene start (GS), at the onset of the gene of L, would be too error-prone. In addition, considering that the CTD-arm is required both for the inter-turn interactions, responsible for the non-canonical NC organisation, and for the prevention of the premature RNA encapsidation by N⁰, the mechanisms behind the strong inhibition of the HRSV RNA polymerase activity by mutations and truncation of the CTD-arm[30] are certainly convoluted.

Along with the considerations summarised in the results section, the structures of the $N_{10}$ double ring and the ring-capped helices and the functional relevance of the observed NTD-NTD interaction in the HRSV RNA synthesis as inferred from the minigenome assay may prove valuable in the light of the recent cryo-ET studies visualising ring-like structures inside the virion[23] and in sub-compartments of the HRSV assembly granules[40]. Indeed, subtomogram averaging analysis of these assemblies is yet to come and neither their exact symmetry nor the identity of the encapsidated RNA are known.

Finally, it is essential to keep in mind that *in cellula* the HRSV RNA synthesis occurs in virally induced cytoplasmic inclusions considered as active viral factories and formed by liquid-liquid phase separation[41–43]. In line with the structural polymorphism of the purified NCs obtained by heterologous expression described here, we hypothesise that particular functional states of the NCs can be enriched in VFs depending on the progress of the viral cycle or the status of certain cellular pathways. The material properties of biomolecular condensates may also influence the NC structures. Thus, in future, it is essential to combine cryo-EM investigation of the structure-function relationships of the HRSV synthesis machinery in vitro with its cryo-ET analysis in the cellular context.

## Methods
### Plasmids and baculoviruses
The codon-optimised sequence coding for the wild type (WT) N (strain Long) was synthesised (GenScript) and cloned in the pFastBacDual vector under the control of the polyhedrin promoter at BamHI and SalI sites. A stop codon was inserted after amino acid residue 370 by site-directed mutagenesis using Q5 Site-Directed Mutagenesis Kit

(NEB, catalogue number E0552S), in order to express the truncated N1-370 construct. Recombinant baculoviruses were recovered using the Bac-to-Bac baculovirus expression system (Invitrogen). N WT or N1-370 bacmids were obtained after transformation of DH10EMBacY bacteria (Geneva Biotech). Recombinant baculoviruses were recovered after transfection of High Five cells using Cellfectin reagent (ThermoFisher Scientific) and amplification.

The plasmid pGEX-PCT used for bacterial expression of the recombinant GST-PCT was obtained by BamHI digestion of the PCR-amplified DNA coding for the C-terminal 161–241 residues of the HRSV P (strain Long) and insertion at the BamHI/SmaI sites into the pGEX-4T3 vector[44] (GE Healthcare Life Sciences). Plasmids for minigenome assay expressing HRSV N, P, M2-1, and L are designated pN, pP, pM2-1 and pL, were obtained from Dr Jin[45]. The M-luciferase subgenomic replicon contains two transcription units, the second encoding the firefly luciferase (*luc*) gene under the control of the M/SH intragenic "Gene Start" sequence. It was engineered by inserting the firefly *luc* at the KpnI-XhoI restriction sites in the pM/SH vector in place of the SH coding region[46] yielding pM/Luc. The plasmids encoding N mutants pNH100E, pNH100E-R101D, pNH100E-E122R, and pN1-370 were generated using Q5 Site-Directed Mutagenesis Kit (NEB, catalogue number E0552S). Primers are shown in Supplementary Table 1.

### Protein expression and purification
Recombinant GST-PCT was used for purification of recombinant N expressed in insect cells. Briefly, *Escherichia coli* BL21 (DE3) bacteria (Novagen) transformed with the pGEX-PCT plasmid were grown at 37 °C, in 2xYT medium with 100 μg/ml ampicillin. After 7 h, an equal volume of 2xYT medium with 100 μg/ml ampicillin and 80 μg/ml of isopropyl β-d-1-thiogalactopyranoside (IPTG) was added to induce protein expression, before overnight incubation of the culture at 28 °C. Bacteria were harvested by centrifugation at 3000 × g for 30 min at 4 °C, resuspended for 30 min in lysis buffer (Tris 50 mM, NaCl 60 mM, EDTA 1 mM, 0.1% Triton X-100, DTT 2 mM, pH 7.8, anti-proteases (Roche)), and sonicated on ice. Benzonase was then added to the lysate, followed by a 30 min incubation at room temperature. After centrifugation at 10,000 × g for 30 min at 4 °C, the supernatant was incubated with Glutathione-Sepharose 4B beads (Cytiva, reference 17075604) for 3 h at 4 °C. The beads were washed once in the lysis buffer and twice in PBS 1× buffer. Beads were resuspended in PBS 1× and stored at 4 °C.

For expression of N, High Five cells (ThermoFisher Scientific, catalogue number B85502) were infected at a multiplicity of infection of 2 for 72 h with the baculovirus coding either for the WT HRSV N or the N1-370 construct. Cells were washed in TEN buffer (50 mM Tris-HCl, 10 mM EDTA, 150 mM NaCl, pH 7.4) and centrifuged at 3000 × g for 5 min. The cells were resuspended in 10 ml of lysis buffer (TEN buffer, NP-40 at 0.6% (v/v), anti-proteases/phosphatases (Roche), RNase (200 μg/ml, Invitrogen), DNase (5 units/ml, Promega)) and incubated 40 min at 37 °C. The lysate was clarified by centrifugation at 14,000 × g for 15 min at 4 °C, then incubated for 3 h at 4 °C with the GST-PCT beads previously rinsed in TEN buffer. The GST-PCT beads were then washed once in lysis buffer and twice in TEN buffer and then incubated in TEN buffer in the presence of thrombin (Sigma), for 72 h at 4 °C. The supernatant was collected and concentrated using a column with a MWCO of 100 kDa (Sartorius).

### Minigenome assay
BSRT7/5 cells, a cell line derived from the BHK-21 cells, constitutively expressing the T7 RNA polymerase[47], were used for the minigenome assay. Cells were grown in Dulbecco's modified Eagle's medium (Lonza) with 2 mM L-glutamine, antibiotics and 10% fetal bovine serum. Cells at 90% confluence in 96-well plate were transfected using Lipofectamine 2000 (Invitrogen) according to manufacturer's instructions with the following plasmid mixture: 62.5 ng of pM/Luc, pP

and pN (WT or N mutants), 31.3 ng of p L, 15.6 ng of pM2-1, and 15.6 ng of pRSV-β-galactosidase (Promega) for transfection efficiency normalisation. After 24 h, the cells were lysed in luciferase lysis buffer (30 mM Tris pH 7.9, 10 mM MgCl₂, 1 mM DTT, 1% Triton X-100, and 15% glycerol). After the addition of the substrate (Luciferase assay system, Promega), the luciferase activities were determined for each cell lysate with an Infinite 200 Pro (Tecan, Männedorf, Switzerland) and normalised based on β-galactosidase (β-Gal) expression. Four replicates were carried out and mean values were calculated. The analysis was done using Excel (Microsoft) and Prism 9 (GraphPad).

Expression of WT and mutant N was assessed by Western blot using a rabbit anti-N antiserum (1:2000)[44] and a mouse monoclonal anti-tubulin antibody (1:1000) (Sigma), revealed by incubation with anti-rabbit and anti-mouse antibodies coupled to HRP (1:10,000) (SeraCare). The following antibodies were used for this assay: mouse monoclonal anti-alpha-tubulin antibody (Sigma, product number T6199, clone DM1A, lot 115M4796V), anti-rabbit secondary antibody, peroxidase-labelled (SeraCare, material number 5450-0010, lot 10437708), anti-mouse secondary antibody, peroxidase-labelled (SeraCare, material number 5450-0011, lot 10430730).

### Cryo-EM data collection

3 μL of the purified FL or truncated HRSV NC sample were applied to a glow-discharged R2/1 300 mesh holey carbon copper grid (Quantifoil Micro Tools GmbH) and plunge-frozen in liquid ethane using a Vitrobot Mark IV (FEI) operated at 100% humidity at room temperature. Datasets were recorded at the EM platform of the IBS Grenoble, on a Glacios microscope (Thermo Scientific) equipped with a K2 summit direct electron detector (Gatan) operated in counting mode. Data collection was performed with SerialEM 4.0. A summary of cryo-EM data collection parameters can be found in Supplementary Table 2. Movies were acquired with a total dose of 42 e⁻/Å² and a defocus range of −0.7 to −2.4 μm, at 1.145 Å/pixel at the specimen level. Cryo-EM data on the N1-370 mutant was acquired with using beam-tilt induced image-shift protocol (9 images for each stage movement). Micrographs were manually screened based on the presence of particles, amount of contamination and apparent beam-induced movement, resulting in a total of 11,386 selected micrographs for the FL NCs and 6,312 selected micrographs for N1-370 NCs. A visual inspection of the full-length NC dataset showed the presence of at least four types of assemblies – helical NCs, double-headed NCs, ring-capped NCs and double rings—which were processed separately. Similarly, the helical NCs and the stacks formed by the N1-370 mutant were also processed separately.

### Image analysis of the non-canonical helical FL HRSV NCs

For the helical NCs, an initial manual picking of 800 filaments from a subset of micrographs was performed with EMAN2 2.99 e2helixboxer.py[48] and used to create a training dataset for crYOLO 1.9.3[49], which was subsequently used for the picking on all micrographs, resulting in 97,280 filaments traced. The filament coordinates were then used for particle extraction in RELION 4.0[50] with a binning to a boxsize of 128 pixels (corresponding to 3.04 Å/pixel) and a 15 Å distance between boxes. A total of 1,406,835 helical segments were picked and iteratively classified in RELION, keeping the straightest 2D class averages at each round, which resulted in 544,972 selected segments. All 2D classes had a similar appearance, contrasting with the paramyxoviral-like herringbone pattern and featuring inwards-shifted densities alternating between the left- and the right-hand side of the pattern every ~1.5 turns (or ~100 Å) (Fig. 1b; Supplementary Fig. 6). This real-space observation was the first indication that the helical arrangement of the HRSV NC was different from the one of paramyxoviral NCs. Thus, the sum of PS of the aligned segments corresponding to 27 classes selected based on estimated resolution and number of particles was calculated with RELION and inspected with

bshow[51]. The PS sum showed a similar pattern for all selected classes, again in strong contrast with the PS of paramyxoviral NCs. Specifically, in addition to an expected layer line with a maximum close to the meridian at ~1/70 Å, attributable to the pitch estimated from the class averages in real space, the PS of HRSV NCs exhibited an additional layer line, with a strong maximum on the meridian, at ~1/100 Å. This indicated the existence of a helical periodicity that should correspond to a ~100 Å rise (Supplementary Fig. 6). Subsequent processing steps were carried out in cryoSPARC 3.3.2[52]. The helical twist was determined on the imported binned segment selection whereby multiple helical 3D refinements were run in parallel using as initial symmetry parameters a fixed 100 Å rise and varying the twist from 60 to 180° with a 10° step. The crystal structure of the HRSV N-RNA ring[6] was used to validate the 3D maps after refinement, choose the best one for further analysis and impose the correct handedness. An examination of this intermediate map and of its refined symmetry parameters (~105 Å rise and ~150° twist) enabled us to understand that the helical HRSV NC in our cryo-EM images was, in fact, a right-handed "super-helix", with a asymmetric unit corresponding to a left-handed helix composed of ~16 adjacent N protomers. Since the rise of this super-helix was larger than the initially defined distance between successive segments, a new template-based picking with the projections of this intermediate map was done in cryoSPARC, using 105 Å as a distance between segments. The picking yielded 769,699 helical segments that were cleaned down to 546,489 particles by further 2D classification. Again, none of the 2D classes had a herringbone aspect. This cleaned particle set was further processed in two different ways. First, it was subjected to a heterogeneous refinement using 5 classes, with no symmetry imposed, in order to select the better-resolved and most regular particles. The three most similar classes were combined into a final set of 389,540 segments that were used for a final helical refinement to an average resolution of 6.2 Å (Fourier Shell Correlation (FSC) at 0.143), which was sharpened with a B-factor of −470 Å² for visualisation and rigid body fit of the crystal structure (Fig. 5). This map reflects the non-canonical helical HRSV NC. Second, because of the flexibility of the super-helix, we decided to focus on one asymmetric unit and performed a refinement with a mask enclosing 16 N protomers created from the helical map. This resulted in a final map, with a subsection of five consecutive well-defined protomers at its centre that had an average resolution of 3.5 Å (FSC at 0.143). The five-protomer-subsection map, sharpened with a B-factor of −94 Å², was used for subsequent model building and structural analysis. Finally, in agreement with the absence of herringbone-like 2D class averages, competitive refinement between a canonical symmetry-enforced map (Supplementary Fig. 6) and the non-canonical symmetry maps did not reveal any subset of segments that would match a canonical symmetry. Thus, we concluded that in the experimental setup used in this study, full-length HRSV NCs do not form any canonical helices with one protomer per asymmetric unit.

The variation of the axial shifts between two consecutive protomers and their distances from the helical axis of the non-canonical HRSV NC helix (Fig. 4c) was calculated based on the coordinates of the centres of gravity of each protomer obtained from the rigid body-fitted crystal structure. To estimate the tilt of each protomer relative to the helical axis, two Cα atoms (from G106 and H274) were picked on the long inertia axis and the angle between the vector formed by these two atoms and the helical axis was computed.

### Image analysis of the double-headed helical NCs and ring-capped NCs

Inspection of the cryoSPARC analysis of the 2D classification from the template-based picking of helical segments, allowed us to identify a 2D class average, corresponding to a class of 10,718 particles, that showed features of a double-headed helix (potentially mixed with ring-capped helices). This class was used for template-based picking, giving a set of

176,614 particles that were iteratively cleaned by 2D classification down to 12,699 particles, mostly by removing double-ring side views. From these particles, an ab initio reconstruction was calculated, followed by non-uniform refinement, which gave a ~7 Å resolution map used to create templates to re-pick the entire dataset. The resulting over 200,000 particles were classified down to 52,127 particles, with a mixture of classes showing double-headed features and classes showing ring-capped helices. Because the separation between double-headed helices and ring-capped helices by 2D classification only did not seem entirely satisfactory, we imported the particles into RELION and iteratively performed 3D classifications until obtainment of two stable particle subsets (and one subset discarded as junk particles). This procedure resulted in 22,162 particles of ring-capped helices and 25,338 particles of double-head helices. These were imported back into cryoSPARC and used for final non-uniform refinements using the RELION 3D class averages as initial references. The average resolutions (FSC at 0.143) of the resulting 3D reconstructions of the ring-capped and the double-headed helices were 3.9 Å and 3.8 Å respectively; finally, the maps were respectively sharpened with a B-factor of −35 Å$^2$ and −54 Å$^2$.

In order to establish the correspondence between the protomers in the double-headed NCs and in the non-canonical helical NCs (and thereby to allow the numbering of the protomers in the asymmetric unit of the non-canonical helix as in Fig. 4), we matched each of the 16 rigid body-fitted protomers of the non-canonical NC onto the rigid body-fitted protomers of the double-headed NCs (alignment done on one protomer) and calculated the RMSD between the two structures over five consecutive protomers. The lowest RMSD value (3.39 Å) indicated the register of the N protomers in the double-headed NCs compared to the helical NC.

### Image analysis of the HRSV N-RNA double rings

In cryoSPARC, manual picking of 50 side views of double rings followed by 2D classification was used to create a 2D template for automatic picking, giving 23,704 particles after cleaning by 2D classification. These particles were used for ab initio 3D reconstruction followed by refinement and new particle picking using the refined map projections. Iterative 2D classification gave a set of 57,896 side views. Ring-like top views were not considered because they could potentially correspond to single rings or short helices instead of double rings. The selected side views were cleaned down to 47,212 particles by an additional 3D classification step in order to remove undetected potential C11-symmetric rings or short helices mixed with the C10-symmetric double ring side views. A final non-uniform refinement led to a 3D reconstruction at an average resolution of 2.9 Å (FSC at 0.143), sharpened with a B-factor of −96 Å$^2$.

### Image analysis of the canonical helices formed by the N1-370 mutant

In cryoSPARC, manual picking of 200 helical segments followed by 2D classification was used to prepare templates for the filament tracer job. Two rounds of 2D classification and re-picking were then performed to yield a set of 471,549 helical segments used for initial reconstruction. Different helical symmetries ranging from 9 to 11 subunits per turn with a starting pitch of 60 Å were tested by running multiple helical refinements, and only the refinement starting at 10 subunits per turn gave an interpretable map with visible secondary structures. This map was used to generate 2D projections for a final template-based filament tracer job, which gave, after keeping only the straightest helical segments by iterative 2D classification steps, a final set of 329,706 segments. The last refinement was run with a mask enclosing 30% of the segment length, giving a reconstruction at an average resolution of 4.3 Å (FSC at 0.143), sharpened with a B-factor of −155 Å$^2$. The refined helical parameters−a −36° twist and a 6.58 Å rise−are very similar to the ones calculated from the helical parameters of the non-canonical

helical NC formed by the FL N, assuming one protomer per asymmetric unit (−35.7° twist and 6.58 Å rise).

### Image analysis of the N1-370 mutant stack

A 2D class average corresponding to stacked rings derived from an automatic picking of helical segments in cryoSPARC was used for template-based particle picking, followed by 2D classification and re-picking until a stable subset of 81,918 particles was obtained. Ab initio 3D reconstruction followed by refinement with imposed D10 symmetry as well as defocus refinement gave a final 3D map at an average resolution of 2.8 Å (FSC at 0.143), which was sharpened with a B-factor of −99 Å$^2$.

### Map visualisation, local resolution calculation, model building and refinement

For all final maps, the local resolution was calculated in cryoSPARC. The most interpretable maps were used for further structural analysis. The crystal structure of the N-RNA monomer (PDB: 2WJ8, chain S) was rigid-body-fitted in the maps with Chimera 1.16[53]. Where appropriate, refinement was performed using the Phenix 1.19.2-4158 software package[54] and manual correction in Coot 0.9.6.2[55]. At different processing stages, the structures were inspected with Chimera and bsoft 2.1.3[51], and figures were done using ChimeraX 1.4[56].

### Statistics and reproducibility

Data was collected on independent experiments. Statistics details are presented in the Methods section and in the figure legends where appropriate.

### Reporting summary

Further information on research design is available in the Nature Portfolio Reporting Summary linked to this article.

## Data availability

The coordinates and structure factors generated in this study (Supplementary Table 2) have been deposited in the EM Data Bank (EMDB) and Protein Data Bank (PDB) under accession codes EMD-17031, PDB: 8OOU (double ring), EMD-17030 (non-canonical helical NC), EMD-17035, PDB: 8OP1 (helical subsection), EMD-17036 (double-headed NC), EMD-17037 (ring-capped NC), EMD-17034 (canonical helical NC formed by the N1-370 mutant), and EMD-17038, PDB: 8OP2 (stack formed by the N1-370 mutant). The data underlying Fig. 4c and Supplementary Figs. 5, 8 and 9 generated in this study are provided in the Supplementary Information/Source Data file. Source data are provided with this paper.

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

## Acknowledgements

We thank Julien Sourimant for providing the pFastBac-N plasmid, Guy Schoehn for establishing and managing the IBS cryo-electron microscopy platform and for providing training and support, Lefteris Zarkadas for assistance at the Glacios microscope and Daphna Fenel and Emmanuelle Neumann for assistance at the negative stain EM platform. This work was funded by the Agence Nationale de la Recherche (grant ANR DecRisp ANR-19-CE11-0017-01 to J.F.E. and I.G.). We used the platforms of the Grenoble Instruct-ERIC centre (ISBG; UAR 3518 CNRS-CEA-UGA-EMBL) within the Grenoble Partnership for Structural Biology (PSB), supported by FRISBI (ANR-10-INBS-0005-02) and GRAL, financed within the University Grenoble Alpes graduate school (Ecoles Universitaires de Recherche) CBH-EUR-GS (ANR-17-EURE-0003). The EM facility is supported by the Rhône-Alpes Region, the Fondation Recherche Medicale (FRM), the fonds FEDER and the GIS-Infrastrutures en Biologie Sante et Agronomie (IBISA). L.G. acknowledges the financial support by the ANR (DecRisp ANR-19-CE11-0017-01) and the Fondation pour la Recherche Médicale (FRM, FDT202204015081).

## Author contributions

L.G., M.B.V., D.C., M.G. and I.G. performed experiments, L.G., A.D. and I.G. analysed the cryo-EM data. L.G., J.F.E and M.G. analysed the biological data. I.G. and J.F.E. designed the overall study. I.G., A.D., J.F.E. and M.G. supervised the project. I.G. wrote the manuscript with contributions from L.G., A.D., M.G. and J.F.E. L.G., A.D. and I.G. prepared the figures. All authors read the manuscript prior to submission.

## Competing interests

The authors declare no competing interests.
