## [Peer Review File · Nature Communications]

Structural landscape of the Respiratory Syncytial Virus nucleocapsidsREVIEWER COMMENTS

Reviewer #1 (Remarks to the Author):

General comments:

The manuscript by Gonnin et al. worked on purified nucleoproteins (N) of human Respiratory Syncytial virus (RSV) and unveiled a series of ensembles including helical nucleocapsids (NC), double-headed nucleocapsids, ring-capped NCs, and double N-RNA rings via cryo-electron microscopy. Based on all structures with descent resolutions, authors clarified the underlying molecular mechanisms for RSV N to assemble into diverse NCs. All their findings highlight the polymorphism of the NCs in RSV and other viruses. Though most ensembles of NCs have been explicitly reported in NDV, SeV, NiV, and others, the co-occurrence of all these ensembles from one virus still has significance to delineate the structural base especially for RSV RNA synthesis.

Specific concerns:

1. Among all these RSV ensembles, the non-canonical helical organization of the RSV NC is of great interest. The asymmetric unit takes a curved shape, and is supposed relevant to the exposure of RNA during RNA synthesis. Besides non-canonical helical NCs, is there any canonical helical nucleocapsids from wild-type N? Meanwhile, it is necessary to verify the function of these non-canonical helices. Indeed, shortening of CTD-arms will transform the non-canonical helix into canonical nucleocapsids. At least, the authors should perform the minigenome assay and compare the activities of wild-type non-canonical capsids and canonical capsids from truncated arm mutants.
2. After the truncation of CTD-arms, N can assemble into canonical helical NC or top-to-top N10 ring stacks. From the shown micrographs in Fig. 6 a and d, it seems that these two kinds of structures are not compatible. Is there any condition or trick to control the assembly of these two structures?
3. Compared with double N-RNA rings, the assembly of N10 ring stacks needs one extra interface. To avoid the possible artifact such as His tags, which might incur oligomerization, authors should clarify which kind of tag is fused to N, and whether tags have been removed before structural analyses. Furthermore, authors need to provide the SDS-page gel of purified N in the manuscript.

4. The polymorphism of RSV NCs is very interesting, while the interface is quite complicated. Authors should delineate the relationship among different structures and mark the interfaces in one protomer to show the potentials for higher-ensembles.

Minor concern:

1. There is a typo in Line 169. It should be “~1.5 turns left-handed”.

Reviewer #2 (Remarks to the Author):

Manuscript by Gonnin et al. describes the cryoEM structure of RSV nucleocapsid and the authors present evidence that the NC is polymorphic. This is a technically sound study that further supports previous work in filoviruses and paramyxoviruses. The key finding, based on the manuscript is the polymorphic nature of the NC and the observation of non-canonical helical organization of the RSV NC. Overall, there's a lot of structural information that can be useful to better understand RSV nucleocapsid, RSV RNA synthesis and virus structure. However, the manuscript does not appear to address any functional questions. Therefore, in the opinion of the reviewer, this work is observational and the findings are premature. Structural studies, particularly comparisons with short and long N protein sequences, relevance of the N C-terminus in engaging other viral and host factors, as well as the significance of the change in NC pitch must be accompanied by validations. Without such functional studies, this manuscript is too premature.

Reviewer #3 (Remarks to the Author):

Mononegaviruses form a helical nucleocapsid that is a minimal unit for viral RNA synthesis. The helical nucleoprotein (N)-genomic RNA complex is the nucleocapsid core and serves as a scaffold for nucleocapsid assembly and RNA synthesis. Therefore, determining the detailed structure of the N-RNA complex in its native helical form is important for a better understanding of the structural basis of viral assembly. Thus far, the helical structure of the human respiratory syncytial virus (HRSV) remains unclear and the N-RNA complex structure has only been determined in a decameric ring-like form. In this study, the authors analyzed the N-RNA complex using single-particle cryo-EM and showed the existence of various

complex structures, including a helical structure with a non-canonical helical organization. Unfortunately, the authors were unable to determine the high-resolution structure of the helical complex which would have allowed for a more detailed investigation of molecular interactions such as inter-strand interactions in the helix.

The identification of polymorphic N-RNA complex structures is intriguing, because it may reflect a particular step in the helical assembly process and potentially contain critical interactions required for the helical assembly. However, the authors conducted little functional analyses using structure-based NP mutants and did not provide biological significance of the respective complexes they observed in this study. Therefore, the manuscript is lacking in terms of its virological aspects. Furthermore, it appears that the difference between the findings of this study and previous studies has not been adequately scrutinized.

Major points:

1. The explanation of the structure and data processing needs to be more straightforward.
2. The authors need to make appropriate reference to previous studies on structural polymorphism of HRSV N-complexes to deeply discuss the differences. For example, Bhella et al. (<https://doi.org/10.1099/0022-1317-83-8-1831>) and MacLellan et al. (<https://doi.org/10.1128/JVI.00526-07>) observed monodisperse 10-subunit (and 11-subunit) N rings using the same system (an insect cell expression system). In addition, Conley et al. (Ref 19) showed that the ring structures were observed in virions budding from virus-infected cells. The reviewer is wondering, for example, why the double-ring and ring-capped complexes identified in the present study were not observed in previous studies, why the 10-subunit ring is absent in the present study, and whether the concentration of the sample had an impact on the polymorphism of the complex.
3. The reviewer suggests that the authors should provide more detailed information and figures related to the image analysis, particularly for helical reconstruction, to support the validity of their findings regarding the non-canonical helix. For example, the reviewer concerns that, despite the presence of some distinguishable secondary structures in the 6.2 Å map provided by the authors, few α -helix-like blobs are visible, raising the possibility that

the structure may have been reconstructed with incorrect helical parameters.

4. Extended Figure 5: The reviewer suggests that it would be interesting for the authors to examine the structures of the mutated N-RNA complexes. For example, were the double-ring and ring-capped complexes formed in these mutants? Has the composition of the complexes been altered due to the mutations?

Minor points:

1. The reviewer suggests that the authors consider to use the latest ICTV virus name abbreviations, e.g., change "RSV" to "HRSV", "hMPV" to "HMPV", "EV" to "EBOV", "MaV" to "MARV", "MeV" to "MV", "SeV" to "SenV", and "Newcastle disease (NDV)" to "avian paramyxovirus 1 (APMV-1)". The others can be found at

https://ictv.global/news/vmr_release_0423.

2. Each chain of the model for the short subsection of helices (Helice-small-turn.pdb) appears to be identical, but there are regions of the map where the model does not fit, especially the C-terminus. The map is an asymmetric reconstruction, and each chain of the model should be refined within the experimental data (map).

3. Figure 3: Labeling panel (a) as "cryo-EM" might lead to confusion, since panels (d) and (e) also display cryo-EM structures. To avoid potential misunderstandings, the authors could consider using more specific and distinctive labels for each panel.

4. Page 18, line 662. The (f) should be (h).

REVIEWER COMMENTS

Reviewer #1 (Remarks to the Author):

General comments:

The manuscript by Gonin et al. worked on purified nucleoproteins (N) of human Respiratory Syncytial virus (RSV) and unveiled a series of ensembles including helical nucleocapsids (NC), double-headed nucleocapsids, ring-capped NCs, and double N-RNA rings via cryo-electron microscopy. Based on all structures with descent resolutions, authors clarified the underlying molecular mechanisms for RSV N to assemble into diverse NCs. All their findings highlight the polymorphism of the NCs in RSV and other viruses. Though most ensembles of NCs have been explicitly reported in NDV, SeV, NiV, and others, the co-occurrence of all these ensembles from one virus still has significance to delineate the structural base especially for RSV RNA synthesis.

We would like to specify that to our knowledge double-headed and ring-capped filaments have only been reported for *Paramyxoviridae* but not for *Pneumoviridae*, and double rings have not been observed in *Paramyxoviridae*. Moreover, the structure of the helical RSV nucleocapsids solved in our study is very different from what was expected based on *Paramyxoviridae* structures, and therefore a careful comparison of all RSV NC assemblies with their paramyxoviral counterparts was a must.

Specific concerns:

1. Among all these RSV ensembles, the non-canonical helical organization of the RSV NC is of great interest. The asymmetric unit takes a curved shape, and is supposed relevant to the exposure of RNA during RNA synthesis. Besides non-canonical helical NCs, is there any canonical helical nucleocapsids from wild-type N?

We thank the reviewer for this very relevant comment. Indeed, we extensively explored our data for a presence of a fraction of canonical helices which would have a pitch of $\sim 70\text{\AA}$ and around 10 subunits per turn. However, at least in our experimental setup, the full-length RSV NCs do indeed adopt only a non-canonical symmetry. This was already apparent at the level of the 2D classes, none of which had a herringbone aspect, and at the level of the PS, all featuring an additional layer line with a strong maximum on the meridian, at $\sim 1/100\text{\AA}$. In addition, at the 3D reconstruction level, none of our attempts to identify a small subpopulation of segments consistent with a canonical symmetry, yielded a reasonable reconstruction. We now modify the manuscript to clearly explain this point as follows. Lines 146-150: "Although at first glance, the 2D class averages of the RSV NCs with a continuous filament course suggest a paramyxoviral-like arrangement with a herringbone appearance and a $\sim 70\text{\AA}$ pitch, their careful scrutiny shows that in all class averages every ~ 1.5 turns (or $\sim 100\text{\AA}$) densities at either the left- or the right-hand side of the pattern are shifted inwards (Figure 1b; Supplementary Figure 6)." Lines 419-424: "All 2D classes had a similar appearance, contrasting with the paramyxoviral-like herringbone pattern and featuring inwards-shifted densities..." Line 444: "Again, none of the 2D classes had a herringbone aspect." Lines 456-461: "Finally, in agreement with the absence of herringbone-like 2D class averages, competitive refinement between a canonical symmetry-enforced map (Supplementary Figure 6) and the non-canonical symmetry map did not reveal any subset of segments that would match a canonical symmetry. Thus, we concluded that in the experimental setup used in this study, full-length RSV NCs do not form any significant amount of canonical helices with one protomer per asymmetric unit."

Meanwhile, it is necessary to verify the function of these non-canonical helices. Indeed, shortening of CTD-arms will transform the non-canonical helix into canonical nucleocapsids. At least, the authors should perform the minigenome assay and compare the activities of wild-type non-canonical capsids and canonical capsids from truncated arm mutants.

We fully agree with this comment. An extensive minigenome analysis of CTD-arm mutants had been described in the previous work by two of the co-authors of this manuscript, JFE and MG (Esneau et al., 2019), who showed that many point mutations of the CTD-arm resulted in a strong reduction of the polymerase activity. Therefore, for the initial submission, we assumed that the N1-370 truncation mutant should be deficient in polymerase activity. However, we anticipated a potential criticism and during the

revision created a plasmid expressing RSV N1-370 for the minigenome assay and performed the experiment. As expected, the CTD-arm truncation completely abolished the polymerase activity. We now include this information in Supplementary Figure 8 and describe our findings in lines 220-225 “As expected from previous studies³⁰, the CTD-arm truncation completely abolishes the polymerase activity in an RSV minigenome assay (Supplementary Figure 8), highlighting the importance of the CTD-arm of N in the RSV RNA synthesis. In spite of the truncation, the N1-370 mutant retained its interaction with P and therefore the corresponding N-RNA assemblies could be purified following the same protocol as the one used for the full-length (FL) construct and analysed by cryo-EM”.

2. After the truncation of CTD-arms, N can assemble into canonical helical NC or top-to-top N10 ring stacks. From the shown micrographs in Fig. 6 a and d, it seems that these two kinds of structures are not compatible. Is there any condition or trick to control the assembly of these two structures?

As the Reviewer, we were also surprised to discover that the truncation of the CTD-arms resulted not only in canonical helices but also in stacks of NTD-NTD and CTD-CTD assembled decameric rings. These structures coexist in the same preparation, but for an unknown reason on the cryo-EM grid they tend to group by type in the grid holes. Therefore, we have images that mostly feature canonical helices (like in the Figure 6a that also shows many top views of single rings, one side view of a single ring and some side views of NTD-NTD and CTD-CTD double rings), and images that mostly feature stacks (like in the original Figure 6d that also showed some short canonical helices at the left part, as well as top views of single rings and some side views of CTD-CTD double rings). In addition, in some images the co-occurrence of helices and stacks is more prominent. Thus, to avoid misunderstanding, we have now replaced the panel 6d with a representative micrograph in which helices, stacks and rings are clearly visible at the same time. We have not attempted to control the assembly into one or the other structure but rather see their coexistence as an additional manifestation of the remarkable polymorphism of the RSV NCs and the importance of the CTD-arm for the assembly and flexibility of the NCs.

3. Compared with double N-RNA rings, the assembly of N10 ring stacks needs one extra interface. To avoid the possible artifact such as His tags, which might incur oligomerization, authors should clarify which kind of tag is fused to N, and whether tags have been removed before structural analyses. Furthermore, authors need to provide the SDS-page gel of purified N in the manuscript.

We agree with the reviewer - in addition to the NTD-NTD interaction required for the bottom-to-bottom double formation, the assembly of N₁₀ rings into stacks also involves the CTD-CTD interface and is mediated by a massive reorientation of the truncated CTD-arm and a Y365-Y365 stacking, as we describe in the manuscript. As specified in the Methods section, all our work has been performed on a WT construct of N, without any tag, and using affinity purification with a C-terminal construct of the RSV phosphoprotein P (residues 161-241) fused to GST. Thus, there is no reason for an artifactual assembly to take place. As requested, we now provide an SDS page of purified N (that also demonstrates the absence of P-CTD in the final NC sample used for cryo-EM) as a Supplementary Figure 9.

4. The polymorphism of RSV NCs is very interesting, while the interface is quite complicated. Authors should delineate the relationship among different structures and mark the interfaces in one protomer to show the potentials for higher-ensembles.

We thank the reviewer for this suggestion and add a supplementary figure 10 that schematically depicts the involvement of specific structural elements of N in different lateral and longitudinal interactions underlying the remarkable polymorphism of HRSV NC-like assemblies.

Minor concern:

1. There is a typo in Line 169. It should be “~1.5 turns left-handed”.

Done

Reviewer #2 (Remarks to the Author):

Manuscript by Gonnin et al. describes the cryoEM structure of RSV nucleocapsid and the authors present evidence that the NC is polymorphic. This is a technically sound study that further supports previous work in filoviruses and paramyxoviruses. The key finding, based on the manuscript is the polymorphic nature of the NC and the observation of non-canonical helical organization of the RSV NC. Overall, there's a lot of structural information that can be useful to better understand RSV nucleocapsid, RSV RNA synthesis and virus structure. However, the manuscript does not appear to address any functional questions. Therefore, in the opinion of the reviewer, this work is observational and the findings are premature. Structural studies, particularly comparisons with short and long N protein sequences, relevance of the N C-terminus in engaging other viral and host factors, as well as the significance of the change in NC pitch must be accompanied by validations. Without such functional studies, this manuscript is too premature.

We thank the reviewer for acknowledgement of the technical soundness of our work. We would like to emphasize that in our view this study does not merely support previous work on filoviruses and paramyxoviruses but rather reveals the unique features of pneumoviral nucleocapsids and extensively discuss their potential functional implications. It therefore perfectly fulfils the primary aim of any structural biology manuscript by providing new directions for functional studies and stimulating creative ideas that can now be addressed by the virology community, who will build their further experiments on our pioneering research.

Reviewer #3 (Remarks to the Author):

Mononegaviruses form a helical nucleocapsid that is a minimal unit for viral RNA synthesis. The helical nucleoprotein (N)-genomic RNA complex is the nucleocapsid core and serves as a scaffold for nucleocapsid assembly and RNA synthesis. Therefore, determining the detailed structure of the N-RNA complex in its native helical form is important for a better understanding of the structural basis of viral assembly. Thus far, the helical structure of the human respiratory syncytial virus (HRSV) remains unclear and the N-RNA complex structure has only been determined in a decameric ring-like form. In this study, the authors analyzed the N-RNA complex using single-particle cryo-EM and showed the existence of various complex structures, including a helical structure with a non-canonical helical organization. Unfortunately, the authors were unable to determine the high-resolution structure of the helical complex which would have allowed for a more detailed investigation of molecular interactions such as inter-strand interactions in the helix.

As previously shown by others, and as extensively demonstrated and discussed in our work, the helical RNA NCs do not have a long-range order. This property is inherent to all currently observed *in vitro* RSV NCs, no matter if they were obtained by purification out of the virion or the infected cells, or by heterologous expression of N. Noteworthy, RSV NCs may potentially assume more ordered conformations under certain cellular conditions. In the present case however, even at the level of one asymmetric unit, composed of 16 N protomers, the inherent absence of the long-range order of the filament makes them to a particularly difficult target for high resolution analysis, in spite of a large number of particles and extensive classification performed to select the most regular filaments only. A 6.2 Å average resolution of the obtained map is already a major achievement because it reveals the unprecedented non-canonical symmetry of the RSV NC. It allows visualisation of alpha-helices and an unambiguous fit of the N monomer, and shows a clear and continuous RNA density. Moreover, to go even further in the analysis and assess the short-range order of the NC filament, we refined a short helical subsection from the centre of the asymmetric unit to 3.5 Å average resolution. While this subsection is shorter than one helical turn and therefore cannot offer insights into the longitudinal interactions, it is still showing the variation of the protomer tilt, thus validating the structure of the full-length NC helix.

The identification of polymorphic N-RNA complex structures is intriguing, because it may reflect a particular step in the helical assembly process and potentially contain critical interactions required for the helical assembly. However, the authors conducted little functional analyses using structure-based NP mutants and did not provide biological significance of the respective complexes they observed in this study. Therefore, the manuscript is lacking in terms of its virological aspects.

In the first version of the manuscript, we analysed the functional relevance of the NTD-NTD interface observed in the double-ring, ring-capped and double-helix structures, using an RSV minigenome

assay. This allowed us to show that one of the two mutations resulted in a circa 90% reduction of the polymerase activity, which suggests a possible functional role of the observed NTD-NTD interactions in the RSV RNA synthesis. To address the Reviewer's comment and also to follow the suggestion of Reviewer 1, in this revised version we now also show that truncation of the last 21 residues of the CTD-arm in the N1-370 construct completely abolishes the polymerase activity in an RSV minigenome assay, which means, in agreement with the previously published studies by Esneau et al., 2019, that these C-terminal residues of N are important for RSV RNA synthesis. Furthermore, the discovery that this mutant forms rigid canonical helices instead of highly curved non-canonical ones alone is already a structure-based functional validation as such because this mutant was designed based on our structure of the non-canonical NC in which the CTD-arm is involved in inter-turn interactions. Moreover, in contrast to this mutant, the FL RSV NCs feature systematic variations in the RNA accessibility, that should have immediate implications for viral synthesis. Finally, all our structural observations are carefully discussed in the light of the available functional data in the discussion sections and we propose different functional hypotheses and structure-based research directions for further functional studies.

Furthermore, it appears that the difference between the findings of this study and previous studies has not been adequately scrutinized.

We kindly disagree with the reviewer on this point. Indeed, to our knowledge no high resolution cryo-EM studies on helical pneumoviridae NCs has so far been published. Furthermore, in our manuscript we make it a point of honour to extensively compare our structures with all available high resolution cryo-EM structures of *Paramyxoviridae* and *Filoviridae* NCs (see also the answer below to the point 2).

Major points:

1. The explanation of the structure and data processing needs to be more straightforward.

We agree with the Reviewer that the non-canonical structure of the helical RSV NC with 16 differently tilted and shifted N subunits in a 1.5-turn asymmetric unit is very unusual (the only other known example being the Dahlemense TMV model which we describe in the discussion section). This is why the entire "Cryo-EM analysis reveals a non-canonical symmetry of the helical RSV NC" chapter of the result section is deliberately more technical than the rest of the main text and dedicated to the strategy used for structure solving and interpretation. Additional technical details are provided in the corresponding Methods section. Thanks to the Reviewer comment, we now realise that to avoid repetition some steps of our reasoning were explained in more details in the main text than in the Method section. We have now introduced even more explanations in the Methods section, for example (lines 419-431) "All 2D classes had a similar appearance, contrasting with the paramyxoviral-like herringbone pattern and featuring inwards-shifted densities alternating between the left- and the right-hand side of the pattern every ~ 1.5 turns (or ~ 100 Å) (Figure 1b; Supplementary Figure 6). This real-space observation was the first indication that the helical arrangement of the RSV NC was different from the one of paramyxoviral NCs. Thus, the sum of PS of the aligned segments corresponding to 27 classes selected based on estimated resolution and number of particles was calculated with RELION and inspected with bshow⁵³. The PS sum showed a similar pattern for all selected classes, again in strong contrast with the PS of paramyxoviral NCs. Specifically, in addition to an expected layer line with the maximum close to the meridian at $\sim 1/70$ Å, attributable to the pitch estimated from the class averages in real space, the PS of RSV NCs exhibited an additional layer line, with a strong maximum on the meridian, at $\sim 1/100$ Å. This indicated the existence of a helical periodicity that should correspond to a ~ 100 Å rise (Supplementary Figure 6)." and "Finally, in agreement with the absence of herringbone-like 2D class averages, competitive refinement between a canonical symmetry-enforced map (Supplementary Figure 6) and the non-canonical symmetry maps did not reveal any subset of segments that would match a canonical symmetry. Thus, we concluded that in the experimental setup used in this study, full-length RSV NCs do not form any significant amount of canonical helices with one protomer per asymmetric unit." (lines 456-461). These additions also address concerns of the Reviewer 1.

We are convinced that together the dedicated results and methods sections, the Figures 4 and 5, and especially the Supplementary Figure 6, explain the image analysis strategy and the resulting non-canonical helical assembly in great detail. We hope that the readers interested in technical aspects of helical image processing will particularly appreciate the Supplementary Figure 6 and its detailed legend.

Finally, to explain the non-canonical structure from a less technical and more visual perspective, we now provide a Supplementary Movie 1 that shows a morph between the non-canonical FL RSV NC and the

canonical N1-370 RSV NC thereby allowing to visually grasp the differences between the two assemblies and the huge structural rearrangements involved.

2. The authors need to make appropriate reference to previous studies on structural polymorphism of HRSV N-complexes to deeply discuss the differences. For example, Bhella et al. (<https://doi.org/10.1099/0022-1317-83-8-1831>) and MacLellan et al. (<https://doi.org/10.1128/JVI.00526-07>) observed monodisperse 10-subunit (and 11-subunit) N rings using the same system (an insect cell expression system).

In the original version of the manuscript, we chose to cite the cryo-ET (Liljeroos et al., 2013 and Conley et al., 2021) and X-ray crystallography (Tawar et al., 2009) studies because they contained 3D structural information on the helical nucleocapsid and not only on the ring-like assemblies. In addition, we thoroughly discussed the negative stain ET study of Bakker et al., 2013 because, in spite of its extremely low resolution, the authors could combine the negative stain ET map with the crystal structure to produce a first model of the helical NC, and notice the tripartite interaction inside the N-hole. To satisfy the Reviewer's criticism, we now also include references to morphological 2D observations and 3D analysis by negative stain and cryo-negative stain EM. Lines 59-61: "Earlier negative stain and cryo-negative stain EM studies reported polymorphism of HRSV N-RNA assemblies observed as flexible filaments and rings^{6,19,20} upon virion lysis or by heterologous expression of N".

In addition, Conley et al. (Ref 19) showed that the ring structures were observed in virions budding from virus-infected cells. The reviewer is wondering, for example, why the double-ring and ring-capped complexes identified in the present study were not observed in previous studies, why the 10-subunit ring is absent in the present study, and whether the concentration of the sample had an impact on the polymorphism of the complex.

We thank the Reviewer for this important remark. Indeed, the C10 and C11 rings are also present in our study (see for instance Figure 1a, Figure 6a, d), we simply decided not to aim for their 3D reconstruction, firstly, because, as typical for such assemblies, they have a strongly preferred top view orientation and, secondly, because they have already been widely studied by others. Thus, we instead chose to focus on the D10 symmetric double ring that has not yet been described. The revised version now contains the following (lines 66-70): "In addition to the well known C10- and C11-symmetric rings^{6,19,20} with a strongly predominant top view orientation, a significant amount of side views corresponding to two decameric rings stacked bottom-to-bottom could be detected. Since this assembly, hereafter termed N₁₀ double ring, had not been previously described, its 3D cryo-EM map was derived from the corresponding side view classes."

The information present in the manuscripts describing the low resolution studies done by other groups does not allow us to infer why the NTD-NTD double ring structures have not been previously observed. This might be for instance due to the differences in the sample purification (CsCl gradient versus GST-PCT affinity purification). The sample concentration used in cryo-negative stain and in cryo-EM experiments is similar and therefore cannot really be an issue.

Interestingly, as described in our manuscript, NTD-NTD double rings are also present in the RSV and hMPV crystal structures.

Finally, while ring-like shapes are observed in virions and in RSV assembly granules, their structures have not yet been determined and their exact symmetry or symmetries (C10, C11, D10, D11, short spirals, etc) is unknown. We now add the following lines (321-327) : «Along with the considerations summarised in the results section, the structures of the N₁₀ double ring and the ring-capped helices, and the functional relevance of the observed NTD-NTD interaction in the RSV RNA synthesis as inferred from the minigenome assay may prove valuable in the light of the recent cryo-ET studies visualising ring-like structures inside the virion²³ and in sub-compartments of the RSV assembly granules⁴⁰. Indeed, subtomogram averaging analysis of these assemblies is yet to come and neither their exact symmetry nor the identity of the encapsidated RNA are known."

3. The reviewer suggests that the authors should provide more detailed information and figures related to the image analysis, particularly for helical reconstruction, to support the validity of their findings regarding the non-canonical helix. For example, the reviewer concerns that, despite the presence of some distinguishable secondary structures in the 6.2 Å map provided by the authors, few α -helix-like blobs are visible, raising the possibility that the structure may have been reconstructed with incorrect helical parameters.

Several answers to this point are provided above and in the answer to Reviewer 1. Strong arguments for the correctly determined helical symmetry are listed in the dedicated results and methods sections, and in Supplementary Figure 6. In particular, the obtained non-canonical helical structure of FL RSV NCs explains the experimentally observed 2D class averages and the PS. Furthermore, the symmetry is validated by the masked refinement of a single asymmetric unit, resulting in a well-defined map of five consecutive protomers at the centre of the asymmetric unit with an average resolution of 3.5 Å (Figure 1g; Supplementary Figure 1e, h). Importantly, the variation of the protomer tilt and shift is visible even in this short five protomer-subsection of the helix. Thus, the tilt and shift variation is an inherent characteristics of the RSV NCs and the determined helical symmetry is correct. The reason why the resolution of the asymmetric unit deteriorates towards the mask periphery is a progressive loss of regularity due to a short-range order of the RSV NC helix.

We would like to clarify that the 6.2 Å map shown in Figure 5 and eliciting the Reviewer's concern is the one obtained before the masked refinement of the asymmetric unit (EMD:17030). We realise that this was not explicitly stated in the figure legend. We now specify this point in the Methods (lines 447-450: "The three most similar classes were combined into a final set of 389,540 segments that was used for a final helical refinement to an average resolution of 6.2 Å (Fourier Shell Correlation (FSC) at 0.143), which was sharpened with a B-factor of -470 Å² for visualisation and rigid body fit of the crystal structure (Figure 5)" and in the legend to the Figure 5 "Cryo-EM map of the helical NC before further refinement of the asymmetric unit is shown (EMD:17030)". We now also cite the EMDB codes of each map in the legends of the Figure 1 and 6 to avoid misunderstanding.

4. Supplementary Figure 5: The reviewer suggests that it would be interesting for the authors to examine the structures of the mutated N-RNA complexes. For example, were the double-ring and ring-capped complexes formed in these mutants? Has the composition of the complexes been altered due to the mutations?

We thank the reviewer for these suggestions. We fully agree that the H100E-E122R mutant deficient in the minigenome assay and predicted to be unable to form double-headed helices, ring-capped helices and double rings should now be structurally analysed. To this end, while this paper was in revision, we constructed the required recombinant baculovirus, expressed the mutant in insect cells and purified it following the protocol used for the WT and the N1-370 mutant. Negative stain images do not seem to show NTD-NTD interaction but a massive data collection and a meticulous and time-consuming cryo-EM analysis, beyond the scope of the current manuscript, will be necessary to definitely conclude on this issue. We provide a negative stain image of the mutant here for Reviewer's appreciation and would prefer not to include it in the present manuscript but to conduct a thorough cryo-EM study of this and other mutants, currently in preparation, and to publish it independently as a follow-up work, with an additional functional analysis.

Minor points:

1. The reviewer suggests that the authors consider to use the latest ICTV virus name abbreviations, e.g., change "RSV" to "HRSV", "hMPV" to "HMPV", "EV" to "EBOV", "MaV" to "MARV", "MeV" to "MV",

"SeV" to "SenV", and "Newcastle disease (NDV)" to "avian paramyxovirus 1 (APMV-1)". The others can be found at https://ictv.global/news/vmr_release_0423.

We thank the reviewer for this suggestion. Indeed, the paper was submitted before the release of the latest recommendations. We have now modified the text and the figures accordingly.

2. Each chain of the model for the short subsection of helices (Helice-small-turn.pdb) appears to be identical, but there are regions of the map where the model does not fit, especially the C-terminus. The map is an asymmetric reconstruction, and each chain of the model should be refined within the experimental data (map).

We are very grateful to the reviewer for spotting this error – in a hurry to satisfy the editor’s respect and to submit the maps and the models to EMDB and PDB in order for the manuscript to be sent out for review, we mixed up the files and deposited an intermediate model of the helical subsection instead of the final one. We have now re-deposited the correct model refined without NCS.

3. Figure 3: Labeling panel (a) as "cryo-EM" might lead to confusion, since panels (d) and (e) also display cryo-EM structures. To avoid potential misunderstandings, the authors could consider using more specific and distinctive labels for each panel.

We would like to note that the NTD-NTD interface in the double ring, the double-headed NC and the ring capped NC is indeed the same. Therefore, in the original version we simply called it “cryo-EM interface” in contrast with the interface formed by crystal contacts in 2WJ8 and 5FVC. To address this comment, the labels have now been changed to “double ring, solution”, “double ring, crystal”, “double-headed NC, solution”, “ring-capped NC, solution», and “solution/crystal” for the middle panel.

4. Page 18, line 662. The (f) should be (h).

Done

REVIEWER COMMENTS

Reviewer #1 (Remarks to the Author):

The authors have addressed most of my concerns. One more concern is about the reconstruction of “ring-capped NC”. From Figure 1 b,c,d, and e, “ring-capped NC” in HRSV lies between “double-ring” and “double-headed NCs” in structure. The authors mentioned that it is hard to distinguish “ring-capped NC” from “double-headed NC” in cryoSparc in the methods. Is it possible that ring-capped NC is a clam-shaped structure as in NDV and SeV? Are there still many “double-headed NCs”, which are involved in the final reconstruction of “ring-capped NC”? To double check this part, the authors should trace the positions of “ring-capped NC” in the filaments in the raw micrographs. Apparently, “ring-capped NC” should appear only on one end of the filaments.

Another concern from the reviewer is to improve the data quality via providing a new micrograph (Fig. 6d) without ice contamination.

Reviewer #3 (Remarks to the Author):

The reviewer has now carefully read the revised version and appreciates the authors' efforts in clarifying the technical aspects of the analysis. Overall, the manuscript has shown some improvement in terms of clarity in cryo-EM data analysis. However, the reviewer still feels that the novelty from a virological point is lacking and the manuscript is observational, which does not significantly advance our understanding of the virus structure and function and does not generate interest among researchers in diverse fields.

1. Finally, all our structural observations are carefully discussed in the light of the available functional data in the discussion sections and we propose different functional hypotheses and structure-based research directions for further functional studies.

The reviewer cannot evaluate this response as sufficient because the authors cannot clarify the virological significance of the molecular structures they found and cannot rule out the possibility that these complexes are artifacts in this manuscript.

2. We thank the Reviewer for this important remark. Indeed, the C10 and C11 rings are also present in our study (see for instance Figure 1a, Figure 6a, d), we simply decided not to aim for their 3D reconstruction, firstly, because, as typical for such assemblies, they have a strongly preferred top view orientation and, secondly, because they have already been widely studied by others. Thus, we instead chose to focus on the D10 symmetric double ring that has not yet been described. The revised version now contains the following (lines 66-70): “In addition to the well known C10- and C11-symmetric rings 6,19,20 with a strongly predominant top view orientation, a significant amount of side views corresponding to two decameric rings stacked bottom-to-bottom could be detected. Since this assembly, hereafter termed N10 double ring, had not been previously described, its 3D cryo-EM map was derived from the corresponding side view classes.”

The reviewer expresses deep concern about this. It is not scientifically appropriate to fail to mention known facts in the data set or exclude them from quantitative evaluation.

3. We are very grateful to the reviewer for spotting this error – in a hurry to satisfy the editor’s respect and to submit the maps and the models to EMDB and PDB in order for the manuscript to be sent out for review, we mixed up the files and deposited an intermediate model of the helical subsection instead of the final one. We have now re-deposited the correct model refined without NCS.

The file provided by the author (DEPOSITED_RSV_Helice-small-turn.pdb) is the same as before.

REVIEWER COMMENTS

Reviewer #1 (Remarks to the Author):

The authors have addressed most of my concerns. One more concern is about the reconstruction of “ring-capped NC”. From Figure 1 b,c,d, and e, “ring-capped NC” in HRSV lies between “double-ring” and “double-headed NCs” in structure. The authors mentioned that it is hard to distinguish “ring-capped NC” from “double-headed NC” in cryoSPARC in the methods. Is it possible that ring-capped NC is a clam-shaped structure as in NDV and SeV? Are there still many “double-headed NCs”, which are involved in the final reconstruction of “ring-capped NC”? To double check this part, the authors should trace the positions of “ring-capped NC” in the filaments in the raw micrographs. Apparently, “ring-capped NC” should appear only on one end of the filaments.

This Reviewer’s comment makes us realise that our description of the separation of the images corresponding to double-headed and ring-capped helices in the corresponding Methods section might have been confusing. We did not mean to convey an impression that this separation was particularly difficult, our goal was simply to explain that to further improve the separation between these structural states, the initial separation performed at the 2D level (in cryoSPARC) was followed by 3D classification (in RELION). We now modified the corresponding paragraph in the Method section to specify this point unambiguously and write in particular that “The resulting over 200,000 particles were classified down to 52,127 particles, with 2D class averages showing either double-headed or ring-capped helices. To further improve the separation between double-headed helices and ring-capped helices based on 2D classification, we imported the particles into RELION and iteratively performed 3D classifications until obtainment of two stable particle subsets (and one subset discarded as junk particles).” The 2D class averages shown at the Figure 1b and the 3D cryo-EM maps shown in Figure 1c,d,e and Supplementary Figure 1a,b,c clearly demonstrate that this approach led to a successful separation between the double rings, the ring-capped helices and the double-headed helices.

In all modern image processing pipelines positions of each extracted particle are indicated at the raw micrographs and we coined the term “ring-capped helix” precisely because the ring is capping the helix at the barbed end (as shown in Figure 1a), in contrast to the term “double-headed helix” that we adopted to describe two helices associated via their barbed ends. Considering that the data set contains 11,386 micrographs and the final reconstruction of the ring-capped helices contains 22,162 particles, on average only one or two ring-capped helices are present per micrograph. Thus, we think that making a dedicated supplementary figure illustrating this point would not be appropriate. We nevertheless provide a figure below for the Editorial assessment: this figure shows three representative 2D class averages of the ring-capped helices (in addition to the one already provided in Figure 1b) and a set of close-up views of raw micrographs (in addition to the one already provided in Figure 1a).

Figure for editorial appreciation. Upper row: three 2D classes of ring-capped NCs featuring a clear ring-capping of the NC barbed end. Lower row: positions of barbed end ring-capping in raw micrographs.

Another concern from the reviewer is to improve the data quality via providing a new micrograph (Fig. 6d) without ice contamination.

Although the presence of ice contamination does not affect the data quality, to satisfy this concern we now provide another micrograph for Figure 6d. Some contamination is still visible but in our view it simply represents the reality of day-to-day cryo-EM and is not harmful but natural.

Reviewer #3 (Remarks to the Author):

The reviewer has now carefully read the revised version and appreciates the authors' efforts in clarifying the technical aspects of the analysis. Overall, the manuscript has shown some improvement in terms of clarity in cryo-EM data analysis. However, the reviewer still feels that the novelty from a virological point is lacking and the manuscript is observational, which does not significantly advance our understanding of the virus structure and function and does not generate interest among researchers in diverse fields.

1. Finally, all our structural observations are carefully discussed in the light of the available functional data in the discussion sections and we propose different functional hypotheses and structure-based research directions for further functional studies.

The reviewer cannot evaluate this response as sufficient because the authors cannot clarify the virological significance of the molecular structures they found and cannot rule out the possibility that these complexes are artifacts in this manuscript.

We prefer to avoid polemics and leave this point to the Editorial assessment.

2. We thank the Reviewer for this important remark. Indeed, the C10 and C11 rings are also present in our study (see for instance Figure 1a, Figure 6a, d), we simply decided not to aim for their 3D reconstruction, firstly, because, as typical for such assemblies, they have a strongly preferred top view orientation and, secondly, because they have already been widely studied by others. Thus, we instead chose to focus on the D10 symmetric double ring that has not yet been described. The revised version now contains the following (lines 66-70): "In addition to the well known C10- and C11-symmetric rings 6,19,20 with a strongly predominant top view orientation, a significant amount of side views corresponding to two decameric rings stacked bottom-to-bottom could be detected. Since this assembly, hereafter termed N10 double ring, had not been previously described, its 3D cryo-EM map was derived from the corresponding side view classes."

The reviewer expresses deep concern about this. It is not scientifically appropriate to fail to mention known facts in the data set or exclude them from quantitative evaluation.

We have to express our strong disagreement with this statement. We are firmly convinced that our description of our approach is perfectly appropriate. Indeed, we say that "In addition to the well known C10- and C11-symmetric rings with a strongly predominant top view orientation, a significant amount of side views corresponding to two decameric rings stacked bottom-to-bottom could be detected. Since this assembly, hereafter termed N10 double ring, had not been previously described, its 3D cryo-EM map was derived from the corresponding side view classes." We did not fail to mention any data, the top views of rings are clearly visible in the images and mentioned in the text. However, we chose not to focus on their 3D reconstruction because (i) one cannot obtain a high resolution 3D reconstruction from top views and therefore additional cryo-EM grid optimisation to counteract this strongly preferred orientation would have been required, (ii) this huge experimental effort would have been very expensive and unnecessary because the structure of the single ring is known, (iii) the goal of the work presented in the current manuscript is explicitly stated in many places throughout the text – the nucleocapsid state involved in viral RNA replication and transcription is helical, and although several structures of paramyxo- and filoviral nucleocapsids have been solved, the structure of the helical pneumoviral nucleocapsid is unknown. Thus, it is the helical NC structure (and the associated ring-capped and double-helical structure) that is the object of the present study, and not the already described single-ring structure. Therefore, we refuse to modify this part of the manuscript.

We would also like to draw attention to the well-known fact that purification of biological samples and cryo-EM grid preparation is never “perfect”: the object of interest is rarely 100 % pure, particles get broken during purification or upon interaction with the air-water interface during the grid freezing, some sorts of them may be preferentially absorbed to the carbon or segregate in ice regions of specific thickness and not get imaged at all, etc. Thus, dependent on the exact scientific question, it is common practice in the cryo-EM world to select specific subsets of particles for further high resolution analysis and discard others. In the present manuscript, we report five structures of the full-length RSV NCs (and two alternative arrangements formed by a C-terminally truncated nucleoprotein mutant).

3. We are very grateful to the reviewer for spotting this error – in a hurry to satisfy the editor’s respect and to submit the maps and the models to EMDB and PDB in order for the manuscript to be sent out for review, we mixed up the files and deposited an intermediate model of the helical subsection instead of the final one. We have now re-deposited the correct model refined without NCS.

The file provided by the author (DEPOSITED_RSV_Helice-small-turn.pdb) is the same as before.

As soon as the Reviewer informed us that we (accidentally) submitted the intermediate pdb of the helical subsection to the data bank instead of the final one, we immediately asked the EMDB to replace it with the final file. Thus, in the EMDB/PDB the EMD:17035 and PDB:8OP1 for the helical subsection are correct. We uploaded the pdb and the validation report to the article folder for your reference.